# Quantitative analyses of T cell motion in tissue reveals factors driving T cell search in tissues

**David J Torres[1]\*, Paulus Mrass[2], Janie Byrum[2], Arrick Gonzales[1], Dominick N Martinez[1], Evelyn Juarez[1], Emily Thompson[3], Vaiva Vezys[3], Melanie E Moses[4], Judy L Cannon[2,5]\***

[1]Northern New Mexico College, Española, United States; [2]Department of Molecular Genetics and Microbiology, University of New Mexico School of Medicine, Albuquerque, United States; [3]Department of Microbiology and Immunology, University of Minnesota Medical School, Minneapolis, United States; [4]Department of Computer Science, University of New Mexico, Albuquerque, United States; [5]Autophagy, Inflammation, and Metabolism Center of Biomedical Research Excellence, University of New Mexico School of Medicine, Albuquerque, United States

**\*For correspondence:**
davytorres@nnmc.edu (DJT);
JuCannon@salud.unm.edu (JLC)

**Competing interest:** The authors declare that no competing interests exist.

**Abstract** T cells are required to clear infection, and T cell motion plays a role in how quickly a T cell finds its target, from initial naive T cell activation by a dendritic cell to interaction with target cells in infected tissue. To better understand how different tissue environments affect T cell motility, we compared multiple features of T cell motion including speed, persistence, turning angle, directionality, and confinement of T cells moving in multiple murine tissues using microscopy. We quantitatively analyzed naive T cell motility within the lymph node and compared motility parameters with activated CD8 T cells moving within the villi of small intestine and lung under different activation conditions. Our motility analysis found that while the speeds and the overall displacement of T cells vary within all tissues analyzed, T cells in all tissues tended to persist at the same speed. Interestingly, we found that T cells in the lung show a marked population of T cells turning at close to 180°, while T cells in lymph nodes and villi do not exhibit this "reversing" movement. T cells in the lung also showed significantly decreased meandering ratios and increased confinement compared to T cells in lymph nodes and villi. These differences in motility patterns led to a decrease in the total volume scanned by T cells in lung compared to T cells in lymph node and villi. These results suggest that the tissue environment in which T cells move can impact the type of motility and ultimately, the efficiency of T cell search for target cells within specialized tissues such as the lung.

## Editor's evaluation

Your unbiased analysis of T cell motion in different tissue settings has revealed that many parameters of T cell motion are similar across tissues, whereas other parameters, particularly related to confinement, are highly sensitive to the tissue microenvironment. Your agnostic approach, meticulous analysis, and clear results are valuable and the technical approach is convincing in its simplicity. The work will be of interest to scientists working in diverse fields from immunologists tracking white blood cell motion to developmental biologists tracking germ cells in *Drosophila* larvae.

## Introduction

Cell migration is a key feature of cellular function, and T cells are particularly specialized to migrate in different tissue types as infection can occur in any tissue and T cell movement in individual tissues is crucial to clear infection. Prior to infection, naive T cells move within the paracortex of the lymph node (LN), and upon interaction with cognate antigen bearing dendritic cells (*Miller et al., 2004*), T cells activate and effector CD8 T cells move to peripheral tissue sites of infection. In tissue sites, CD8 T cells enter infected tissues and move within tissues in order to find and kill target cells, including virally infected cells or tumor cells. Interestingly, T cells can move through many tissue environments that differ in cell type, structure, and chemical cues such as chemokines each of which may affect T cell motility patterns. While many studies have identified key molecules that regulate CD8 T cell effector function in different tissues, still relatively little quantitative analysis has been done to analyze the way CD8 T cells navigate multiple different types of tissue environments to find target cells.

CD8 T cell motility is a key feature of CD8 T cell function, particularly in searching through complex tissue environments to identify and interact with target cells. Motility of T cells is a function of a combination of T cell-intrinsic mechanisms, the extracellular environment, and chemical signals in the milieu (*Krummel et al., 2016*). Tissue environments include a complex and heterogeneous system of cell types, extracellular matrix components, and soluble factors which have been shown to impact T cell motion; for example, structural cells within the tissue environment provide signals to feed back to immune cells in the central nervous system (*Bartholomäus et al., 2009*; *Tomlin and Piccinini, 2018*). Additionally, multiple studies have shown that naive T cells in the LN paracortex use interactions with fibroblastic reticular cells to mediate movement, as well as receive soluble signals such as IL-7 for survival (*Bajénoff et al., 2006*; *Katakai et al., 2004*; *Link et al., 2007*). In addition to extracellular influences, many studies have defined intrinsic molecular regulators of T cell movement, particularly speed in multiple tissues including LNs (*Banigan et al., 2015*; *Fricke et al., 2016*; *Katakai et al., 2004*; *Gérard et al., 2014*; *Eckert et al., 2019*; *Devi et al., 2021*; *Onder et al., 2012*), skin (*Fowell and Kim, 2021*; *Fernandes et al., 2020*; *Overstreet et al., 2013*; *Ariotti et al., 2015*), Female Reproductive Tract (FRT) (*Beura et al., 2018*), liver (*Guidotti et al., 2015*; *McNamara et al., 2017*; *Rajakaruna et al., 2022*), lung (*Mrass et al., 2017*; *Alon et al., 2021*; *Lambert Emo et al., 2016*) just to name a few. High speeds have been linked to integrins (e.g. LFA-1 and VLA-4) (*Katakai et al., 2013*; *Overstreet et al., 2013*), chemokine receptors (*Asperti-Boursin et al., 2007*; *Okada and Cyster, 2007*; *Ariotti et al., 2015*), as well as signaling molecules such as regulators of the actin cytoskeleton (*Mrass et al., 2017*; *Nombela-Arrieta et al., 2007*; *Petrie and Yamada, 2012*; *Cannon et al., 2013*). These studies identified key molecular drivers and structures that mediate T cell movement within individual tissues, but there remains a gap in analysis to compare how T cell motility patterns might differ between tissues.

Quantitative analysis of cell motion provides a powerful tool to determine underlying mechanisms that drive how cells, including T cells, move. While structures, cell types, and chemical cues may differ in tissues that can impact T cell motility patterns, studies performed both in vitro and in vivo have found that all cell movement, including T cells, use actomyosin contractility and actin flow to couple directional persistence and speed, pointing to a universal mechanism for cells to move faster and more persistently in a direction (*Jerison and Quake, 2020*; *Maiuri et al., 2015*). This universal coupling of directional persistence and speed is most clearly shown in cells moving

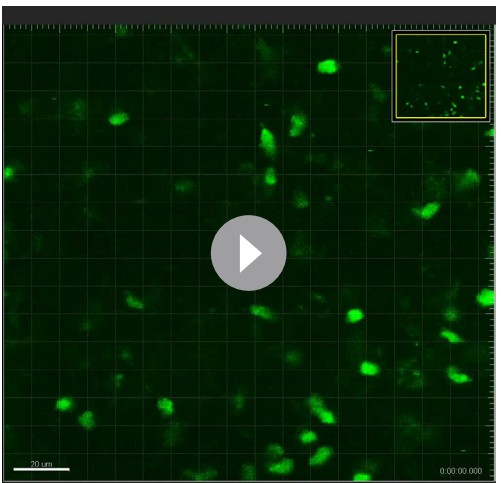

**Video 1.** GFP+ naive T cells moving in lymph nodes (LNs). Naive T cells were isolated from LNs and spleen of Ubiquitin-GFP animals and adoptively transferred into naive C57Bl/6 recipients, then imaged using two-photon microscopy as described in *Fricke et al., 2016*, PLoS Computational Biology. GFP+ naive T cells were imaged as described, tracked, and analyzed. The video contains a representative image from multiple fields of LNs imaged. The data are reproduced under the Creative Commons CC-BY 4.0 license.

https://elifesciences.org/articles/84916/figures#video1

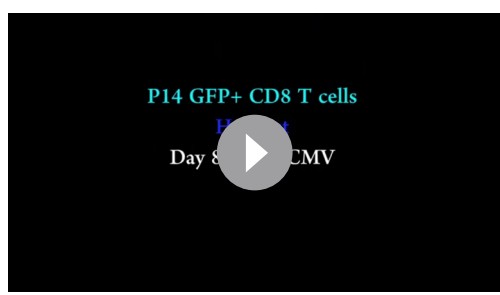

**Video 2.** Migration of CD8 T cells in small intestinal villi at day 8 after lymphocytic choriomeningitis virus (LCMV) infection. Naive P14-GFP CD8 T cells were transferred to B6 mice that were infected with LCMV 1 day later. At days 5 and 8 after infection, the jejunum was imaged via TPLSM. The representative time-lapse videos show P14-GFP CD8 T cells (cyan) at the indicated time points. Hoechst stain (blue) was injected prior to imaging. Reproduced from Thompson et al. Cell Reports 2019 Video S1 under CC BY-NC-ND 4.0 license. Only D8 T cells are shown and D5 movie removed from original file.

https://elifesciences.org/articles/84916/figures#video2

in vitro and on 2D surfaces. How T cells navigate complex tissue environments in three dimensions is still not well understood where cells use multiple modes of migration (*Yamada and Sixt, 2019*).

In this study, we quantitatively analyze T cell movement as one way to interrogate potential environment influences from different tissues. We previously used quantitative analyses of T cell movement in tissue to reveal specific types of motility patterns leading to more effective T cell responses (*Fricke et al., 2016*; *Mrass et al., 2017*; *Thompson et al., 2019*). In this paper, we compare multiple features of T cell motion in different tissues: speed, tendency to persist at a speed, dependence of speed on turning angle, mean squared displacement (MSD), directionality, confined ratio and time, and volume patrolled within the LN with naive T cells and activated CD8 T cells within the small intestine and lung. By comparing T cell movement in different tissues, we identify tissue-specific effects on T cell motility. Our results suggest that tissue environments may contribute to different modes of T cell movement, which can impact the efficiency of T cell searches for target cells in tissues.

## Results
### Speed

We began our analysis with a comparison of the cell-based and displacement speeds of T cells in multiple tissues including naive CD4 and CD8 T cells in the LN in the absence of infection (*Fricke et al., 2016*) (LN) (*Video 1*); effector CD8 T cells moving in the villi in response to lymphocytic choriomeningitis virus infection at day 8 post infection (*Thompson et al., 2019*) (Villi) (*Video 2*); effector CD8 T cells moving in Lipopolysaccharide (LPS)-inflamed lung at days 7–8 post infection (*Mrass et al., 2017*) (Lung LPS) (*Video 3*); and effector CD8 T cells in influenza-infected lung at days 7–8 post infection (Lung Flu) (*Video 4*). Specifics about the cell tracks analyzed for each condition are found in *Table 1*. *Table 2* shows statistics for a reduced data set where outliers are eliminated.

We previously found no difference in motility speed and patterns of naive CD4 and CD8 T cells in LNs (*Fricke et al., 2016*). To ensure consistency across analyses, we normalized time steps to 90 s for all data sets (for details, see Materials and methods). *Figure 1A, B* show the box-and-whisker plot of cell-based speed and displacement speed from each tissue.

The median cell-based speed for naive T cells in the LN was 6.2 µm/min, CD8 effector T cells in the villi 6.5 µm/min, CD8 effector T cells from influenza-infected lung (Flu) 5.2 µm/min and CD8

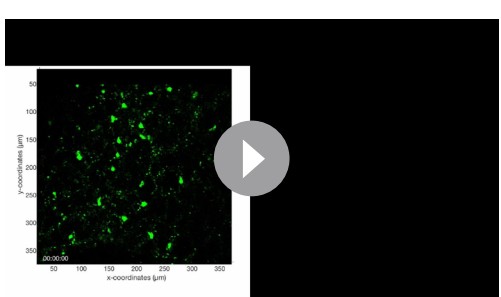

**Video 3.** Migration of CD8+ effector T cells in explanted lungs from LPS-inflamed animals. Left panel: Maximum projection of movie-sequence capturing adoptively transferred T cells (green) within an explanted lung. Trajectories (white lines) show the position of analyzed cells over time. Hours:minutes:seconds are shown in the left bottom corner. Right panel: 3D depiction of cell positions (green circles) and trajectories (blue lines) over time. To improve depth perception, the image volume is rotating during replay. Reproduced from Supplementary Movie 1 in *Mrass et al., 2017*, Nature Communications, Supplementary Movie 1 under Creative Commons CC-BY 4.0 license.

https://elifesciences.org/articles/84916/figures#video3

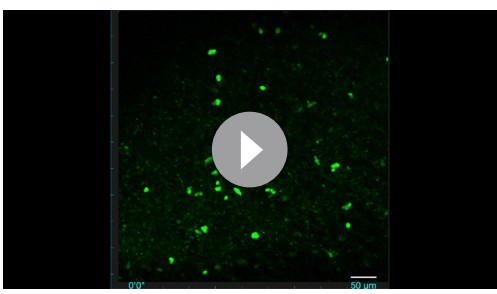

**Video 4.** Migration of CD8+ effector T cells in lungs at d8 after HKx31 influenza infection. Naive CD8 T cells from Ubiquitin-GFP animals were isolated and adoptively transferred into naive C57Bl/6 mice, then infected with $1 \times 10^3$ HKx31. At d8 post infection, lungs were removed and imaged with a heated and oxygenated chamber.

https://elifesciences.org/articles/84916/figures#video4

effector T cells from LPS-inflamed lung (LPS) 4.3 µm/min. Pairwise p-values based on cell-based average speeds from two different tissues are reported in *Table 3* . The p-values are computed using the paired Wilcoxon rank sum test (otherwise known as the Mann–Whitney *U*-test) using the statistical package R with the Bonferroni correction for multiple comparisons. The correction was used for all tables showing p-values.

*Figure 1A* shows that naive T cells in LNs and effector CD8 T cells in the gut villi moved at similar speeds (*Figure 1A*; LN (6.2 µm/min); villi (6.5 µm/min); *Table 3*). Both naive T cells in the LN and effector CD8 T cells in the gut villi moved significantly faster than effector CD8 cells in LPS-inflamed lung (4.3 µm/min) (*Table 3*). Effector CD8 T cells in the influenza-infected lung moved slightly faster than effector CD8 T cells in the LPS-inflamed lung (*Figure 1A*: lung (Flu) 5.2 µm/min versus lung (LPS) 4.3 µm/min ($p = 4.9 \times 10^{-5}$)).

*Figure 1—figure supplement 1A* in the Supplementary Data shows the frequency distribution of cell-based speeds for T cells moving in each tissue type. The figure shows that T cells in either influenza-infected lung or LPS-inflamed lung have a large proportion of cells moving at slower speeds compared to T cells in LNs or villi, with T cells in LPS-inflamed lung showing the largest proportion of cells moving at slow speeds.

We also analyzed effector CD8 T cells moving in the villi at day 5 (d5) post infection and compared with effector T cells moving in villi at day 8 (d8) post infection (Supplementary Data, *Figure 1—figure supplement 2*). A direct comparison of effector T cells moving d5 versus d8 post infection in the villi show that some motility parameters remain similar, including persistence (*Figure 1—figure supplement 2*). Effector T cells in the villi move slightly faster at d8 compared to d5 (*Figure 1—figure supplement 2A, B*), likely reflecting decreasing antigen load with clearance of virus at later times post infection.

The similar speeds of naive T cells in LNs and effector T cells in villi at d8 suggests that activation status is not a sole driver of T cell speed in different tissues despite the significant changes in expression of cell surface markers that regulate motility. Naive T cells express CCR7 and CD62L, while activated T cells upregulate many different cell surface receptors including CD44, CD103, as well as tissue homing chemokine receptors such as CCR9 for gut homing (*Masopust et al., 2010*; *Masopust et al., 2007*; *Olson et al., 2012*; *Wherry et al., 2007*), and CXCR3 and CXCR4 for lung homing (*Fadel et al., 2008*; *Kohlmeier et al., 2009*; *Mikhak et al., 2013*; *Ozga et al., 2022*; *Wein et al., 2019*) . It has been shown that antigen increases interaction time and the difference in speed between effector CD8 T cells on d5 and d8 likely reflect antigen load (*Halle et al., 2016*). However, antigen interaction cannot

**Table 1.** Two-photon microscopy T cell data.

| | Lymph node (LN) | Small intestine (villi) | Lung (Flu) | Lung (LPS) |
|---|---|---|---|---|
| Number of T cells | 4400 | 425 | 355 | 191 |
| Type of T cell | Naive | Activated | Activated | Activated |
| Mode of activation | Not activated | In vivo | In vitro | In vivo |
| T cell specificity | Polyclonal | TCR transgenic | Polyclonal and TCR transgenic | Polyclonal |
| Imaging modality | Tissue explant | In situ | Tissue explant | In situ and tissue explant |
| Source | *Fricke et al., 2016*; *Tasnim et al., 2018* | *Thompson et al., 2019* | Mrass and Cannon | *Mrass et al., 2017* |

**Table 2.** Comparison of inter-tissue versus intra-tissue variability in cell motility.

We performed an analysis of variance (ANOVA) study which computes a ratio of differences in means between groups and within groups. The ANOVA test uses a *F*-distribution which computes a ratio of between- and within-group variance. We previously showed that in the lymph node (LN), motility of T cells captured in different fields and on different days from different LNs of different mice did not contribute significantly to variation within T cell movement of the data set (*Letendre et al., 2015*). To test intra-tissue variability, the groups consisted of frames composed of two-photon tracks within the same tissue of a single mouse. To test inter-tissue variability, the sets consisted of the aggregated frames of two-photon tracks of all mice imaged in the same tissue. To assess intra-tissue variability compared with inter-tissue variability, we performed an ANOVA analysis of cell-based speed. ANOVA analysis shows that while there exist significant differences in T cell motility between the different tissues (Column 3, Row 2), ANOVA analysis also shows that there exist even more significant differences within frames of each individual tissue, particularly of T cells in the LN (Column 3, Rows 3–6). We found the same trend when performing the ANOVA test with the displacement speed and volume per time. To decrease the variability within a tissue type, we selected a reduced set of frames from each tissue based on statistical variability. Twenty-one frames of the most variable frames were removed from the 40 LN frames which increased the ANOVA p-value from $1.7 \times 10^{-204}$ to $4.8 \times 10^{-5}$ when the ANOVA test was re-run with the remaining nineteen frames using the cell-based speed (Row 3, Column 5). Two variable frames were removed from the ten villi frames which increased the ANOVA p-value from $7.7 \times 10^{-19}$ to $8.2 \times 10^{-4}$ (Row 4, Column 5). One frame was removed from the five lung (Flu) frames which increased the ANOVA p-value from $7.6 \times 10^{-7}$ to 0.38 (Row 5, Column 5). No frames were removed from the lung (LPS) data set as none were statistically variable from the other frames. The remaining number of T cells for each tissue is shown in Column 4. The dramatic increase in p-value demonstrates that the removed frames were outliers compared with the data from the same tissue. The data set with outlier frames removed is called the 'reduced data set'. As the variability within each tissue is reduced, the inter-tissue p-value decreased from $3.5 \times 10^{-11}$ to $2.1 \times 10^{-20}$ in Row 2 when the reduced set of files is analyzed. The new inter-tissue p-value $2.1 \times 10^{-20}$ is significantly smaller than the p-values measuring the intra-tissue variability in Column 5, Rows 3–6 .

| | | Complete data set | | Reduced data set | |
| --- | --- | --- | --- | --- | --- |
| | Col 1 | Col 2 | Col 3 | Col 4 | Col 5 |
| Row 1 | Groups | Number of cells and frames | ANOVA p-value | Number of cells and frames | ANOVA p-value |
| Row 2 Inter-tissue variability | Aggregated frames from LN, villi, lung (Flu), lung (LPS) | 5371 | $3.5 \times 10^{-11}$ | 2443 | $2.1 \times 10^{-20}$ |
| Row 3 Intra-tissue | LN frames | 4400 cells 40 frames | $1.7 \times 10^{-204}$ | 1659 cells 19 frames | $4.8 \times 10^{-5}$ |
| Row 4 Intra-tissue | Villi frames | 425 cells 10 frames | $7.7 \times 10^{-19}$ | 296 cells 8 frames | $8.2 \times 10^{-4}$ |
| Row 5 Intra-tissue | Lung (Flu) frames | 355 cells 5 frames | $7.6 \times 10^{-7}$ | 297 cells 4 frames | 0.38 |
| Row 6 Intra-tissue | Lung (LPS) frames | 191 cells 3 frames | $7.9 \times 10^{-3}$ | 191 cells 3 frames | $7.9 \times 10^{-3}$ |

be the sole driver of speed, as antigen-independent CD8 T cells in the LPS-inflamed lung move more slowly than antigen-specific T cells in all other tissues (*Figure 1A, B*). Furthermore, the differences in effector T cell speed between T cells moving in LPS-inflamed lung and influenza-infected lung suggest that the specific tissue environment does not fully dictate the speed of T cell movement.

We also analyzed the intra-tissue variation in T cell motility within each tissue using analysis of variance (ANOVA) (*Table 2*, Column 3, Rows 3–6 shows the p-values). Our results show that within each tissue, particularly the LN, T cells can show high variance in motility. We analyzed motility parameters including cell-based speed when outlier frames with highly variable moving T cells are removed from each tissue (*Table 2*, Column 5, Rows 3–6). Interestingly, while removing variable frames slightly increases cell-based speed and displacement speed (*Figure 1—figure supplement 3*), the relative differences between tissues are preserved (compare *Figure 1* with *Figure 1—figure supplement 3*).

We then calculated the displacement speed of T cells in each tissue, which measures the speed at which the cell moves away from an initial location and is smaller than the cell-based speed in all the tissues (*Figure 1B*). The displacement speed is statistically similar between naive T cells in LN (median 3.7 µm/min) and effector CD8 T cells within the villi (3.3 µm/min). The p-values of comparisons between each T cell type are reported in *Table 4*. We found that the displacement speed of effector CD8 T cells in the lung in both influenza infection and LPS treatment are similar to each other and both statistically significantly lower than T cells in the LNs and villi (influenza-infected lung 1.1

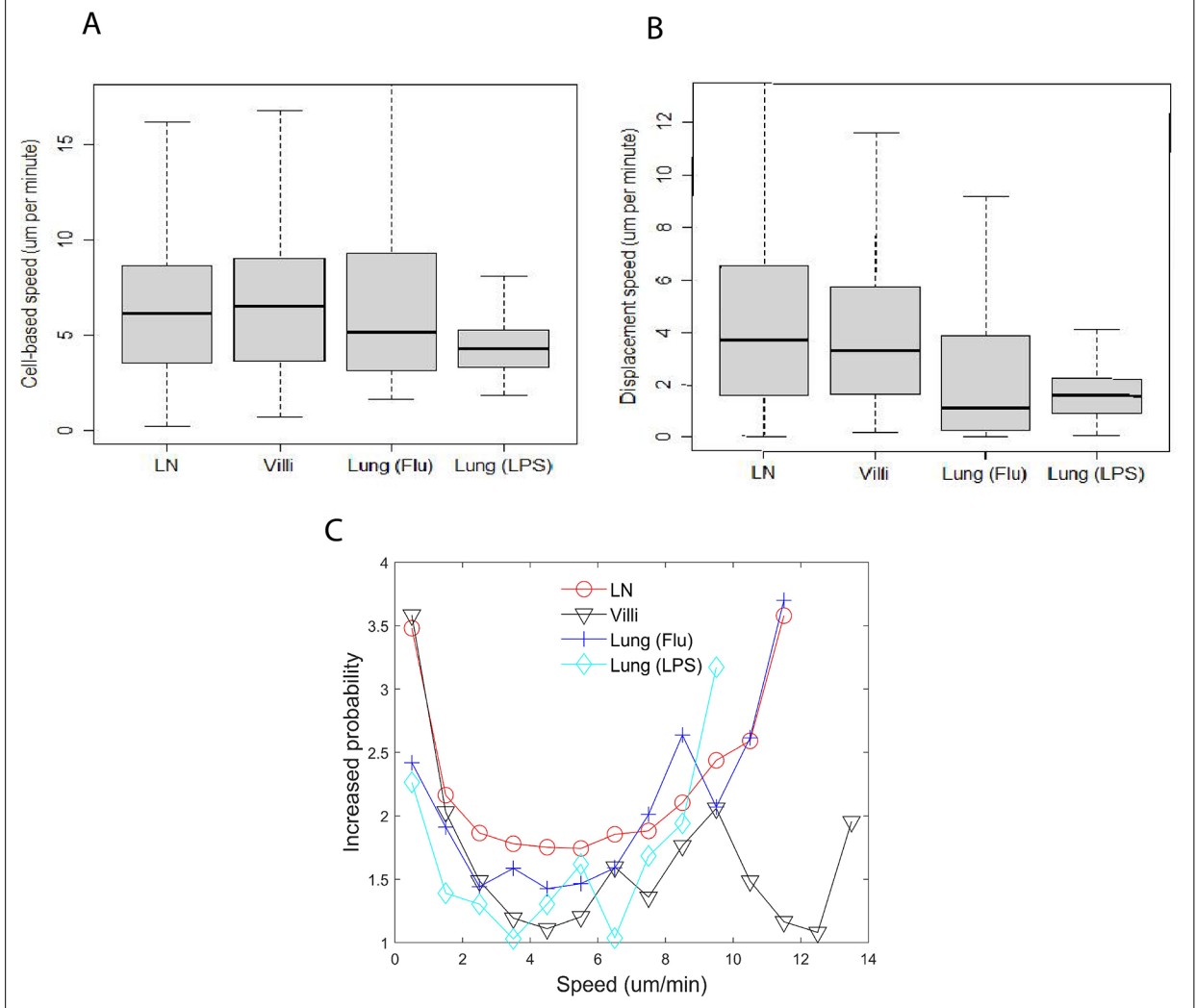

**Figure 1.** Speed distribution of T cells does not correlate with tissue type or activation status. (**A**) Box-and-whisker plot of cell-based speed (µm/min) of T cells moving in lymph node (LN) (median 6.2), villi (median 6.5), lung (Flu infected) (median 5.2), and lung (LPS) (median 4.3). (**B**) Box-and-whisker plot of displacement speed (µm/min) of T cells in lymph (median 3.7), villi (median 3.3), lung (Flu infected) (median 1.1), and lung (LPS instilled) (median 1.6). (**C**) Distribution plot of probability to persist at the same speed.

The online version of this article includes the following source data and figure supplement(s) for figure 1:

**Figure supplement 1.** Speed distribution of cell-based (**A**) and displacement (**B**) speed distributions.

**Figure supplement 2.** Box-and-whisker plot of cell-based speed and displacement speed for villi d5 and villi d8.

**Figure supplement 3.** Reduced data set.

**Figure supplement 3—source data 1.** Editable version of table in *Figure 1—figure supplement 3D*.

**Figure supplement 3—source data 2.** Editable version of table in *Figure 1—figure supplement 3E*.

µm/min, LPS lung 1.6 µm/min). *Figure 1—figure supplement 1B* shows the frequency distribution of T cell displacement speed. These results suggest that the lung environment leads to lower displacement speed of T cells.

## Persistence

We calculated the likelihood that an individual T cell will persist in moving at the same speed in each tissue (*Figure 1C*). We quantified the likelihood that a T cell will continue to move at the same speed as the previous time step and termed this 'persistence' (for detailed methods, see Tendency to persist at a speed). For example, *Figure 1C* shows that for the T cells in the LN, a T cell is 3.5 times more

**Table 3.** Table of p-values of pairwise comparisons of cell-based speed as shown in *Figure 1A* using Wilcoxon rank sum test.

|  | LN | Villi | Lung (Flu) | Lung (LPS) |
|---|---|---|---|---|
| LN | 1.0 | 0.51 | 1.0 | $5.4 \times 10^{-14}$ |
| Villi | 0.51 | 1.0 | 1.0 | $7.8 \times 10^{-14}$ |
| Lung (Flu) | 1.0 | 1.0 | 1.0 | $4.9 \times 10^{-5}$ |
| Lung (LPS) | $5.4 \times 10^{-14}$ | $7.8 \times 10^{-14}$ | $4.9 \times 10^{-5}$ | 1.0 |

likely to continue moving at a slow speed (<1 μm/min) given that it was moving slowly (<1 μm/min) in the previous time step.

We observed a similar trend to persist at very low speeds ($< 2\ \mu m$/min) for T cells moving in all tissues observed and at very high speeds ($> 8\ \mu m$/min) in all the tissues. T cells in the villi exhibited a decrease in persistence at speeds above 10 μm/min while T cells in LN and lung showed similar increase in persistence above 8 μm/min with no decrease at higher speeds. The increased persistence at high speeds in the villi was seen for effector T cells at d5 post infection (*Figure 1—figure supplement 2*). At very low speeds ($< 2\ \mu m$/min), T cells in the LN and villi show a higher likelihood of persistently moving at a slow speed compared to T cells in the lung. *Figure 1C* shows that at intermediate speeds (between $3 - 7\ \mu m$/min), T cells in the lung and villi exhibit lower persistence likelihood than T cells in the LN, suggesting that the lung and gut environment can hinder the ability of T cells to move persistently at these intermediate speeds.

## Mean squared displacement

We then determined the MSD for T cells moving in each tissue. *Figure 2A* plots the MSD for a representative cell versus time from each tissue type. The linear regression line is shown with the scatter plot whose slope is computed using the log of the MSD and the log of time. We then calculated the slope of the linear regression line for each T cell from a tissue. All T cell slopes from a tissue are used to create the box-and-whisker plot shown in *Figure 2B*. T cells are tracked for a maximum of 10.5 min to ensure consistency of analysis across tissues (*Krummel et al., 2016*).

As shown in *Figure 2B*, T cell motion in the LN and villi could be characterized as superdiffusive with values >1. In contrast, the slope of T cells in the LPS-inflamed lung was close to one (0.94) while the slope of T cells in the influenza-infected lung is <1 (0.88) and would be considered diffusive and subdiffusive (*Krummel et al., 2016*). The p-values comparing the differences between the mean square displacement slopes of T cells moving in individual tissues are shown in *Table 5*. The slope of MSD between T cells moving in LN and villi was similar, and significantly different from T cells moving in the lung (*Table 5*). This result remained similar even if outlier frames are removed (*Figure 2—figure supplement 2*).

**Table 4.** Table of p-values of pairwise comparisons of displacement speed shown in *Figure 1B* using Wilcoxon rank sum test.

|  | LN | Villi | Lung (Flu) | Lung (LPS) |
|---|---|---|---|---|
| LN | 1.0 | 1.0 | $< 2 \times 10^{-16}$ | $< 2 \times 10^{-16}$ |
| Villi | 1.0 | 1.0 | $< 2 \times 10^{-16}$ | $< 2 \times 10^{-16}$ |
| Lung (Flu) | $< 2 \times 10^{-16}$ | $< 2 \times 10^{-16}$ | 1.0 | 0.13 |
| Lung (LPS) | $< 2 \times 10^{-16}$ | $< 2 \times 10^{-16}$ | 0.13 | 1.0 |

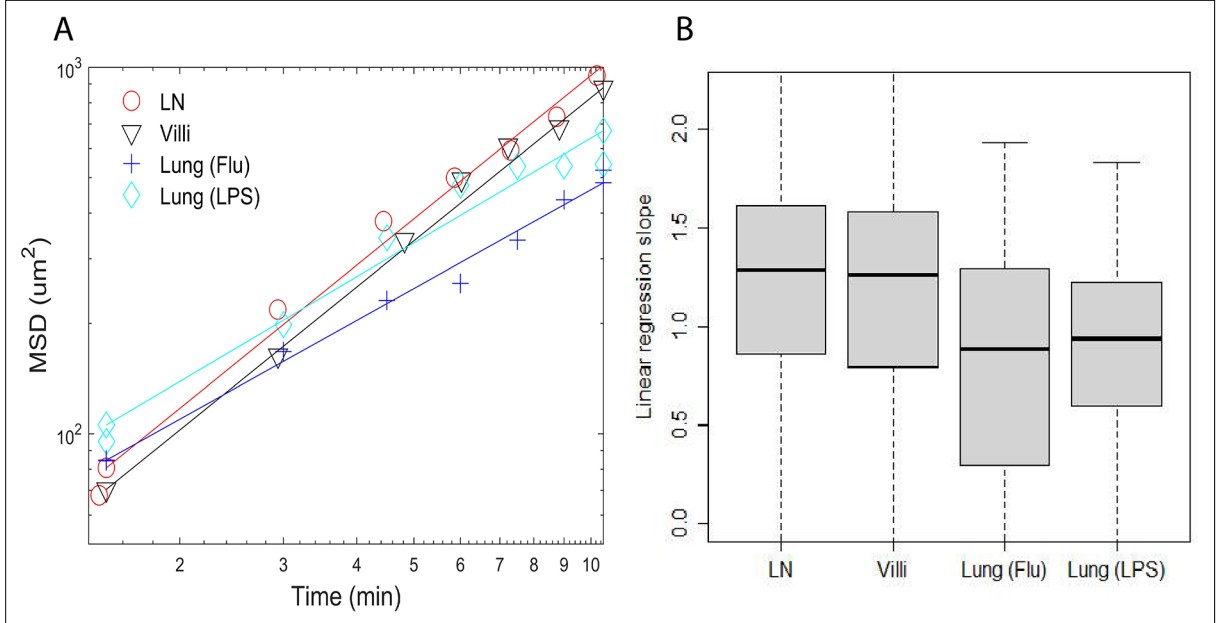

**Figure 2.** Plots of mean square displacement (MSD) versus time and linear regression lines of individual representative cells near median. (**A**) Plots of mean square displacement (MSD) versus time and linear regression lines of individual representative cells near median from **B**. (**B**) Box-and-whisker plots of linear regression cell slopes of log transformed mean squared displacement versus time. The median values are LN (1.3), villi (1.3), lung (Flu) (0.88), and lung (LPS) (0.94).

The online version of this article includes the following source data and figure supplement(s) for figure 2:

**Figure supplement 1.** Mean squared displacement linear regression slope is computed for T cells in the villi at d5 and d8 post infection.

**Figure supplement 2.** Mean squared displacement (MSD) of reduced data set.

**Figure supplement 2—source data 1.** Editable version of table in *Figure 2—figure supplement 2C*.

## Turning angle and dependence of speed on turning angle

As persistence in cell motion is related to turning angles, we analyzed the turning angles of T cells in individual tissues. *Figure 3A* plots the relative frequency of all turning angles of T cells moving in different tissues. We did not include T cells moving at speeds <1 μm/min. We reasoned that turning angles are not relevant when a cell is moving very slowly (< 1 $\mu m$/min). Also small speeds will emphasize turning angles in increments of 45° degrees due to the pixel resolution of the microscope. Since the distribution is not uniform or flat in *Figure 3A*, the cell motion cannot be considered Brownian.

We found that while T cells in all tissues show some preference for turning angles between 40° and 50°, many more T cells in the LN and villi showed the preference to turn at smaller angles compared to T cells in the lung. There was no statistical difference between the LN and villi (see *Table 6* for a list of all the p-values). Interestingly, T cells moving in the lung showed a peak at approximately 160°, a behavior not seen in T cells moving in LN and villi. This peak is likely due to the 'back and forth' motion observed in T cells in the lung which we have previously described (*Mrass et al., 2017*). The higher

**Table 5.** Table showing mean square displacement p-values as pairwise comparisons from *Figure 2B* using Wilcoxon rank sum test.

|  | LN | Villi | Lung (Flu) | Lung (LPS) |
|---|---|---|---|---|
| LN | 1.0 | 0.88 | $< 2.0 \times 10^{-16}$ | $1.8 \times 10^{-15}$ |
| Villi | 0.88 | 1.0 | $1.6 \times 10^{-15}$ | $5.5 \times 10^{-9}$ |
| Lung (Flu) | $< 2.0 \times 10^{-16}$ | $1.6 \times 10^{-15}$ | 1.0 | $9.6 \times 10^{-2}$ |
| Lung (LPS) | $1.8 \times 10^{-15}$ | $5.5 \times 10^{-9}$ | $9.6 \times 10^{-2}$ | 1.0 |

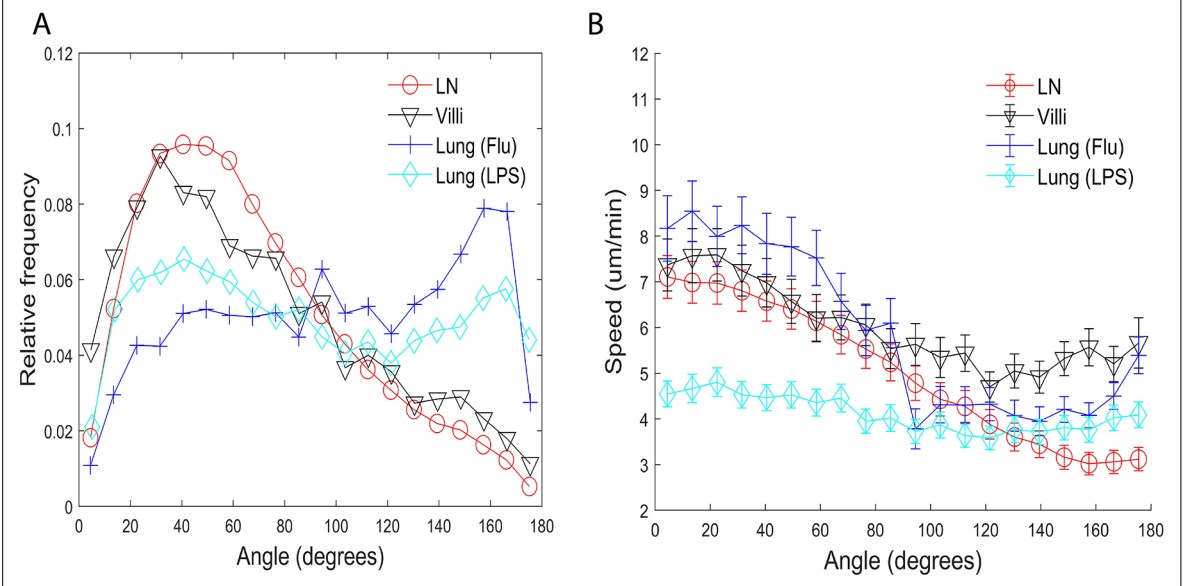

**Figure 3.** Turning angles and coupling of speed and turning angles of T cells in different tissues. (**A**) Relative frequency distribution of turning angles in each tissue. T cells moving in the lung show a peak at approximately 160°. (**B**) Plot of speed (μm/min) versus angle (degrees). The speed tends to decrease as the turning angle increases in all tissues except for T cells in the LPS-inflamed lung. Error bars show plus and minus 1/8 of the standard deviation within each 9° angle bin.

The online version of this article includes the following source data and figure supplement(s) for figure 3:

**Figure supplement 1.** Turning angles and coupling of speed and turning angles of d5 versus d8 CD8 T cells in villi post infection.

**Figure supplement 2.** Turning angles and coupling of speed and turning angles of T cells from the reduced data set.

**Figure supplement 2—source data 1.** Editable version of table in *Figure 3—figure supplement 2C*.

percentage of T cells turning at smaller angles in the LNs and villi suggests that these organs allow for a broader range of turning motion, potentially enabling broader search areas.

We extended our analysis by determining if there exists a relationship between speed and the turning angle. *Maiuri et al., 2015* and *Jerison and Quake, 2020* previously found that T cells that move faster generally move persistently in one direction and show a small turning angle while slower T cells show higher turning angles. Our results confirmed that T cells in the LN, villi, and influenza-infected lung moving with faster speeds exhibit smaller turning angles while T cells moving with slower speeds exhibit larger turning angles for all tissues (*Figure 3B*). Interestingly, effector CD8 T cells moving in the LPS-inflamed lung did not show the speed-turning angle correlation (*Figure 3B*, cyan), suggesting that the relationship between speed and turning angle may not be universal.

The behavior in the LPS lung could be due to the fact that T cells in the LPS-inflamed lung have a slow cell-based speed; however, the flatness of the line suggests that even slow T cells in LPS-inflamed lung may not be subject to the same mechanisms that regulate the speed-angle behavior seen in faster moving cells. We also note that T cells in the influenza-infected lung experience a small increase in speeds for turning angles between 160° and 180°.

**Table 6.** Table of p-values of pairwise comparisons of proportion of turning angles <90° shown in *Figure 3A* using Wilcoxon rank sum test.

|  | LN | Villi | Lung (Flu) | Lung (LPS) |
|---|---|---|---|---|
| LN | 1.0 | 1.0 | $< 2.0 \times 10^{-16}$ | $< 2.0 \times 10^{-16}$ |
| Villi | 1.0 | 1.0 | $< 2.0 \times 10^{-16}$ | $< 2.0 \times 10^{-16}$ |
| Lung (Flu) | $< 2.0 \times 10^{-16}$ | $< 2.0 \times 10^{-16}$ | 1.0 | $2.6 \times 10^{-3}$ |
| Lung (LPS) | $< 2.0 \times 10^{-16}$ | $< 2.0 \times 10^{-16}$ | $2.6 \times 10^{-3}$ | 1.0 |

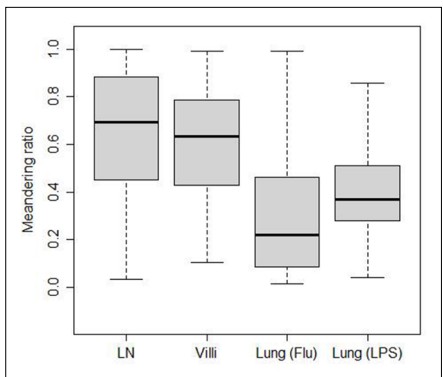

**Figure 4.** Meandering ratio within different tissues. The median values are LN (0.70), villi (0.63), lung (Flu) (0.22), and lung (LPS) (0.37).

The online version of this article includes the following source data and figure supplement(s) for figure 4:

**Figure supplement 1.** Meandering ratio of CD8 T cells within villi of d5 versus d8 post infection.

**Figure supplement 2.** Meandering ratio from the reduced data set.

**Figure supplement 2—source data 1.** Editable version of table in *Figure 4—figure supplement 2B*.

## Directionality and confinement

The turning angle determines the directionality of cell movement. While the displacement speed is lower than the cell-averaged speed in all tissues, the amount of reduction from cell based to displacement speed differs from tissue to tissue. This suggests that directional persistence in T cell movement may differ in the different tissues analyzed. We assessed directionality by calculating a 'meandering ratio' which quantified how likely a T cell deviates from its original direction. *Figure 4* shows the box-and-whisker plot of the meandering ratio of T cells moving within different tissues. T cells in the LN and villi move significantly more directionally than T cells in the lung (median values for the meandering ratio are LN: 0.70, villi: 0.63, lung (Flu): 0.22, and lung (LPS): 0.37). Pairwise p-value comparisons are reported in *Table 7*. The meandering ratio remained the same even after outlier frames were removed (*Figure 4—figure supplement 2*).

We have previously shown that T cells can alternate between confined motion in which the search area is localized and ballistic motion in which motion is fast and persistent in a direction (*Mrass et al., 2017*). We calculated the confined ratio as defined by the time a T cell spends confined versus moving in *Figure 5A*. We found that naive T cells in the LN spend very little time confined and most of the time moving, showing a median confined ratio of 0.15. The confined ratio of effector CD8 T cells in the villi was slightly higher (0.2). Effector CD8 T cells in the influenza-infected lung had a confined ratio of 0.53 while effector T cells in LPS-inflamed lung showed the highest confined ratio of 0.60 (p-values are reported in *Table 8*).

We also calculated the average amount of time T cells from each tissue spend confined which is reported as confined time in *Figure 5B* (p-values reported in *Table 9*). We found that T cells in the lung (Flu) and lung (LPS) showed significantly longer confined times than T cells in LN or villi. Effector T cells in the villi at d5 post infection showed significantly higher confinement ratio and confined time compared with d8 (*Figure 5—figure supplement 1A, B*). Both confined ratios and confined time remained similar even if outlier frames were removed (*Figure 5—figure supplement 2*).

These data show that effector CD8 T cells in LPS-inflamed lung were most confined followed by T cells in influenza-infected lung (Flu) and T cells in the villi. Naive T cells in the LN were the least confined. The low confinement of naive T cells in the LN environment may contribute to the high meandering ratio while confinement as well as antigen in the lung (Flu) likely decrease the ability of T cells to move directionally (*Figure 4*) and lead to low meandering ratios.

## Patrolled volume per time

A key function of T cell movement is surveillance of tissues. To assess whether differences we identified in cell speed, directionality, turning angle, and confined ratio ultimately translate into differences in the ability of T cells to survey tissue, we calculated the volume per time patrolled by a T cell residing in different tissues. Volume per time is a way of incorporating all the different motility parameters we previously identified. *Figure 6A* shows the amount of volume per time patrolled by the T cells in individual tissues. The volume surveyed is highest for naive T cells in LN (median 9.4 $\mu m^3$ /s) and d8 CD8 effector T cells in villi (9.4 $\mu m^3$/s). Effector CD8 T cells in villi at d5 were intermediate at 6.5 $\mu m^3$/s (*Figure 6—figure supplement 1A*). Effector CD8 T cells in the lung showed the lowest volume patrolled: the volume patrolled by T cells in the influenza-infected lung (5.3 $\mu m^3$/s) was statistically similar to the volume patrolled by T cells in the LPS-inflamed lung (5.1 $\mu m^3$/s). See *Table 10* for the p-values.

**Table 7.** Table shows p-values of pairwise comparisons of meandering ratio as shown in *Figure 4* using Wilcoxon rank sum test.

|  | LN | Villi | Lung (Flu) | Lung (LPS) |
|---|---|---|---|---|
| LN | 1.0 | $1.6 \times 10^{-4}$ | $< 2 \times 10^{-16}$ | $< 2 \times 10^{-16}$ |
| Villi | $1.6 \times 10^{-4}$ | 1.0 | $< 2 \times 10^{-16}$ | $< 2 \times 10^{-16}$ |
| Lung (Flu) | $< 2 \times 10^{-16}$ | $< 2 \times 10^{-16}$ | 1.0 | $4.3 \times 10^{-9}$ |
| Lung (LPS) | $< 2 \times 10^{-16}$ | $< 2 \times 10^{-16}$ | $4.3 \times 10^{-9}$ | 1.0 |

We also analyzed the full distribution of volume patrolled by T cells in each individual tissue (*Figure 6B*). The full distribution showed that T cells in LNs and villi at d8 post infection show the largest volume patrolled per time, with T cells in villi at d5 post infection showing a similar distribution of volume scanned (*Figure 6—figure supplement 1B*). Interestingly, although the median patrolled volume is similar between T cells in the LPS-inflamed lung (LPS) and T cells in influenza-infected lung (Flu), T cells in the influenza-infected lung actually show a large number of cells patrolling at both low and large volumes while T cells in LPS-inflamed lung mostly show low patrol volumes. The Kolmogorov–Smirnov test compares the distributions and shows statistically significant differences in all pairwise comparisons (*Table 11*). The results were similar with outlier frames removed (*Figure 6—figure supplement 2A, B*). These data demonstrate that the combination of speed, turning angle, directional movement, and confinement times all contribute to the ability of T cells to search tissue environments for potential targets.

## Discussion

Cell movement through tissue is an important feature in immunity, particularly for T cell-mediated immunity as T cells must make direct cell–cell contact with target cells. A distinguishing feature of T cells is their ability to move through many different tissue environments, which include varying cell

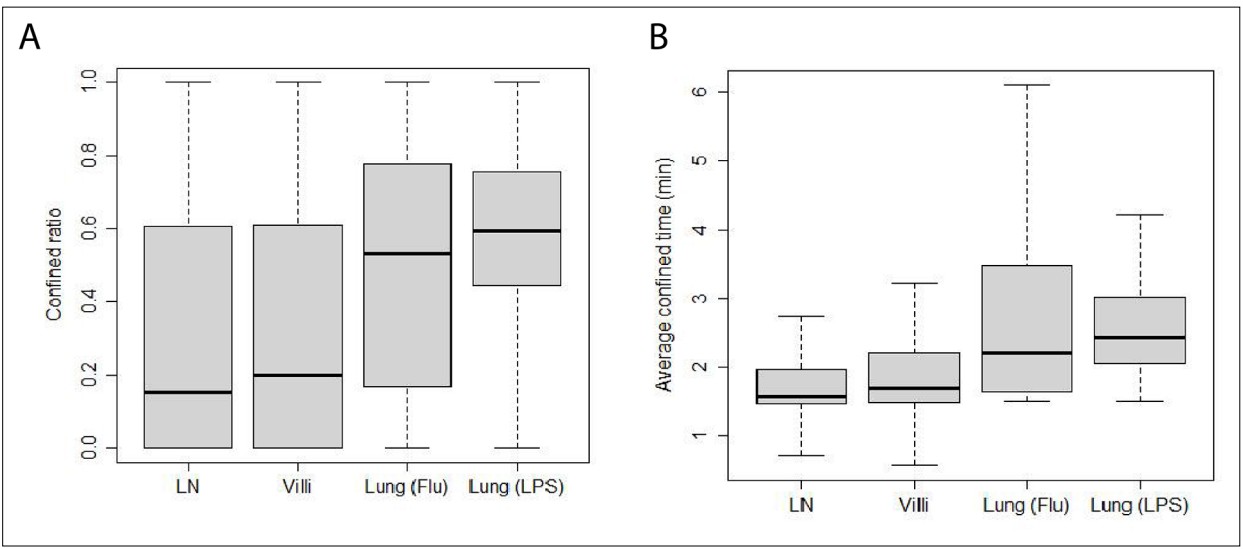

**Figure 5.** Confinement of T cells from different tissues. (**A**) Box-and-whisker plot of confined ratios. Median values: LN 0.15, villi 0.2, lung (Flu) 0.53, and lung (LPS) 0.60. (**B**) Box-and-whisker plot of confined time. Median values (min): LN 1.6, villi 1.7, lung (Flu) 2.2, and lung (LPS) 2.4.

The online version of this article includes the following source data and figure supplement(s) for figure 5:

**Figure supplement 1.** Confinement of T cells from villi d5 versus d8 post infection.

**Figure supplement 2.** Confinement of T cells from the reduced data set.

**Figure supplement 2—source data 1.** Editable version of table in *Figure 5—figure supplement 2D*.

**Figure supplement 2—source data 2.** Editable version of table in *Figure 5—figure supplement 2C*.

**Table 8.** Table showing p-values of pairwise comparisons of confined ratios from *Figure 5A* using Wilcoxon rank sum test.

| | LN | Villi | Lung (Flu) | Lung (LPS) |
|---|---|---|---|---|
| LN | 1.0 | 1.0 | $< 2 \times 10^{-16}$ | $< 2 \times 10^{-16}$ |
| Villi | 1.0 | 1.0 | $1.3 \times 10^{-11}$ | $< 2 \times 10^{-16}$ |
| Lung (Flu) | $< 2 \times 10^{-16}$ | $1.3 \times 10^{-11}$ | 1.0 | $1.8 \times 10^{-2}$ |
| Lung (LPS) | $< 2 \times 10^{-16}$ | $< 2 \times 10^{-16}$ | $1.8 \times 10^{-2}$ | 1.0 |

types, structural features, and chemokines, all of which may contribute to differences in T cell motility (*Donovan and Lythe, 2012*). T cell activation also leads to significant changes in expression of cell surface proteins as well as cytoskeletal machinery that impact T cell motility. Naive T cells express CCR7 and CD62L (*Asperti-Boursin et al., 2007*; *Förster et al., 2008*; *Worbs et al., 2007*). Upon activation, effector T cells upregulate different chemokine receptors depending on the tissue the effector T cells home to as well as integrins and CD44 (*Fadel et al., 2008*; *Kohlmeier et al., 2009*; *Masopust et al., 2010*; *Masopust et al., 2007*; *Mikhak et al., 2013*; *Olson et al., 2012*; *Ozga et al., 2022*; *Thompson et al., 2019*; *Wein et al., 2019*; *Wherry et al., 2007*). Despite these differences, we can use motility parameters to compare the types of movement T cells take independently of the specific molecular interactions that drive these movement patterns.

We analyzed motility parameters to determine what key factors drive the ability of T cells to move through tissues to effectively mount immune responses. Our analysis more fully captures key features of movement that enable T cells to effectively move in tissues, promoting T cell responses. Our study included naive CD8 and CD4 T cells in the LN, antigen-specific effector CD8 T cells responding to infection in the villi and lung (influenza-infected lung), and non-antigen specifically activated effector CD8 T cells in the LPS-inflamed lung in a model of acute lung injury (*Mrass et al., 2017*).

We identified similarities and differences that contribute to the ability of T cells to search tissue for target cells. Interestingly, T cells tend to exhibit similar persistence trends in a range of speeds in all tissue types regardless of activation status. In regards to cell-based speed, non-antigen-specific naive T cells in LNs and antigen-specific effector CD8 T cells in villi move fastest while effector CD8 T cells that are non-antigen specific move slowest in the LPS-inflamed lung. These data suggest that the speed at which T cells move can be independent of antigen-specific interactions as well as activation status (*Bousso et al., 2002*). These results are supported by previous work showing that effector CD8 T cells in the skin appear to move slowly (*Ariotti et al., 2015*) while effector CD8 T cells in the female reproductive tract move at speeds similar to naive T cells in LNs (*Beura et al., 2018*).

Previous work quantitating cell motility in both non-T cells and T cells observed a 'universal coupling' between speed and directional persistence, showing that fast moving cells show directional persistence while slow moving cells do not (*Jerison and Quake, 2020*; *Maiuri et al., 2015*). In our analysis, we find that T cells moving in LN, small intestine villi, and influenza-infected lung all show this coupling. However, we have also identified an exception to this rule for T cells moving in the LPS-inflamed lung, which showed no change in directional persistence as measured by turning angle between fast moving and slow moving cells (*Figure 3B*). Thus, while our data confirm that while speed can be coupled to directional persistence for T cells moving in all tissues analyzed, this coupling is not

**Table 9.** Table showing p-values of pairwise comparisons of confined time from *Figure 5B* using Wilcoxon rank sum test.

| | LN | Villi | Lung (Flu) | Lung (LPS) |
|---|---|---|---|---|
| LN | 1.0 | $2.2 \times 10^{-3}$ | $< 2 \times 10^{-16}$ | $< 2 \times 10^{-16}$ |
| Villi | $2.2 \times 10^{-3}$ | 1.0 | $< 2 \times 10^{-16}$ | $< 2 \times 10^{-16}$ |
| Lung (Flu) | $< 2 \times 10^{-16}$ | $< 2 \times 10^{-16}$ | 1.0 | $5.4 \times 10^{-2}$ |
| Lung (LPS) | $< 2 \times 10^{-16}$ | $< 2 \times 10^{-16}$ | $5.4 \times 10^{-2}$ | 1.0 |

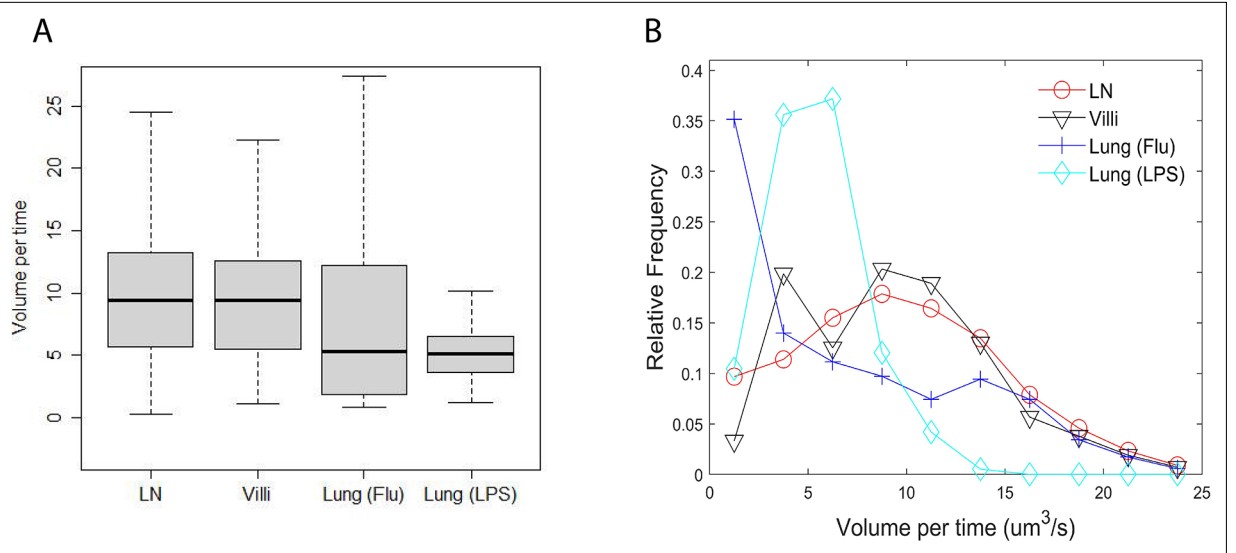

**Figure 6.** Volume patrolled by T cells in different tissues. (**A**) Box-and-whisker plot of median volume per time (μm³/s) patrolled by T cells in LN (9.4), villi (9.4), lung (Flu) (5.3), and lung (LPS) (5.1). (**B**) Relative frequency distribution of volume per time (μm³/s) patrolled by T cells in each tissue.

The online version of this article includes the following figure supplement(s) for figure 6:

**Figure supplement 1.** Volume patrolled by d5 versus d8 T cells in villi post infection.

**Figure supplement 2.** Volume patrolled by T cells in the reduced data set.

necessarily 'universal'. Recently, it has been shown that the correlation between speed and turning angle can arise from differences in sampling rates (*Ganusov et al., 2023*). Our data are unlikely to be affected by sampling rate as we equalized and normalized the sampling rate for all the T cells (see Materials and methods).

We find that T cells moving in the lung show specific motility features that differ from T cells moving in LNs or villi. T cells in the lung tend to displace less, turn at higher angles, particularly at angles >140°, meander more, and linger at locations longer. Confinement can occur in the lung independent of antigen, as CD8 T cells activated in vitro in the LPS-inflamed lung can still show confinement without specific antigen activation. In particular, T cells in the lung exhibit back and forth motion, with a peak in turning angles occurring near 160°. This peak in turning angle is consistent with the 'back and forth' motion we previously observed for T cells in the LPS-inflamed lung, as well as a stop-and-go motion (*Mrass et al., 2017*). Stop-and-go behavior has also been observed albeit for shorter time periods for T cells in LNs by *Miller et al., 2003*; *Wei et al., 2003*, and *Beltman et al., 2007*. We also find that the slope of the MSD versus time for T cells moving in LNs and villi show superdiffusive motion as previously observed for effector T cells in the brain (*Harris et al., 2012*). In contrast, the slope of T cells moving in lung is slightly <1, suggesting Brownian type motion. However, because the angle distribution is not uniform, the cell motion cannot be considered strictly Brownian. Together these motility parameters suggest that the lung environment may lead to specific types of motion taken by T cells, potentially due to the particular physical environment of the lung.

**Table 10.** Table of p-values of pairwise comparisons of volume per time from *Figure 6A* using Wilcoxon rank sum test.

|  | LN | Villi | Lung (Flu) | Lung (LPS) |
|---|---|---|---|---|
| LN | 1.0 | 1.0 | $< 2 \times 10^{-16}$ | $< 2 \times 10^{-16}$ |
| Villi | 1.0 | 1.0 | $6.9 \times 10^{-12}$ | $< 2 \times 10^{-16}$ |
| Lung (Flu) | $< 2 \times 10^{-16}$ | $6.9 \times 10^{-12}$ | 1.0 | 1.0 |
| Lung (LPS) | $< 2 \times 10^{-16}$ | $< 2 \times 10^{-16}$ | 1.0 | 1.0 |

**Table 11.** Table of p-values of pairwise comparisons of volume per time distributions from *Figure 6B* using Kolmogorov–Smirnov test.

|  | LN | Villi | Lung (Flu) | Lung (LPS) |
|---|---|---|---|---|
| LN | 1.0 | 0.07 | $< 2.2 \times 10^{-16}$ | $< 2.2 \times 10^{-16}$ |
| Villi | 0.07 | 1.0 | $< 2.2 \times 10^{-16}$ | $< 2.2 \times 10^{-16}$ |
| Lung (Flu) | $< 2.2 \times 10^{-16}$ | $< 2.2 \times 10^{-16}$ | 1.0 | $3.7 \times 10^{-9}$ |
| Lung (LPS) | $< 2.2 \times 10^{-16}$ | $< 2.2 \times 10^{-16}$ | $3.7 \times 10^{-9}$ | 1.0 |

Our results also show that there can be significant variability of T cell movement patterns within an individual tissue (*Table 2*). However, despite large differences amongst individual T cell movements within any specific tissue, the overall patterns of T cell motility remain similar within tissue and across tissues.

The volume patrolled by a T cell is dependent not only on its speed but also on turning angles, the cell's tendency to meander, and the amount of time the cell spends confined to a location. The higher percentage of T cells turning at smaller angles in the LNs and villi suggests that these organs allow a broader range of turning motion, potentially enabling broader search areas as reflected in the larger volumes patrolled. We found that naive T cells and CD8 effector cells in the villi at d8 post infection patrol a larger volume (9.4 $um^3$/s) due to a combination of fast speed, less confinement, more superdiffusive motion, and smaller turning angles. Analysis of effector T cells at days 5 and 8 post infection in the gut suggest that antigen abundance likely increases confinement and thus decrease T cell speed, leading to a smaller volume patrolled at d5 post infection despite comparable values in meandering ratio and MSD. The patrol volume is significantly smaller for effector T cells in the lung due to greater confinement, more Brownian-like motion, and a greater proportion of large turning angles, particularly angles that suggest a 'back and forth' motion. However, T cells in the influenza-infected lung and LPS-inflamed lung show differences in the frequency distribution of volume per time, with T cells in the influenza-infected lung showing more T cells patrolling at higher volumes than T cells in LPS-inflamed lung. These results suggest that a combination of back and forth motion and slightly faster cell-based speeds can affect the search efficiency for T cells in the influenza-infected lung. Previous results using computational modeling suggests that intermittent and back and forth motion can improve search times (*Bénichou et al., 2011*). Effector T cells moving in LPS-infected lung show the lowest volume covered (5.1 $um^3$/s) due to low speeds, higher turning angles, low directionality and high confinement times. The lack of coupling between speed and turning angle may also lead to low search efficiency.

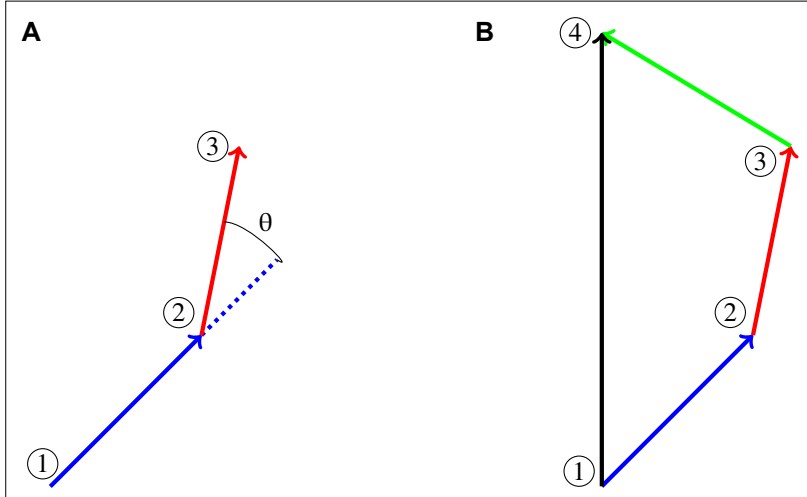

**Figure 7.** Illustration of turning and meandering ratio. (**A**) Turning angle illustration. (**B**) Meandering ratio illustration.

Three-dimensional migration of T cells is a complex interplay of internal cell signaling, the surrounding extracellular tissue environment, molecular signaling and chemokines. We have quantitatively analyzed how T cells move in different tissues using multiple metrics. These metrics provide a way of quantitatively capturing underlying complex features of three-dimensional T cell movement.

## Materials and methods

*Table 1* summarizes the number of T cells tracked within the LN, small intestine villi, and lung which were analyzed. Cell tracks were obtained from at least two separate fields from at least two independent experiments. The data come in the form of *x-*, *y-*, and *z*-coordinates for each T cell at different time frames.

The LN T cell tracks were obtained from data in *Fricke et al., 2016* and *Tasnim et al., 2018*. *Video 1* in the Supplementary Data shows a representative movie from one field from data analyzed in *Fricke et al., 2016*. The T cell tracking in the small intestine was previously described in *Thompson et al., 2019*. *Video 2* reproduces day 8 (d8) T cell movement in villi from Video S1 shown in *Thompson et al., 2019*. T cells from LPS-inflamed lung were previously described in *Mrass et al., 2017*. *Video 3* shows CD8 T cells moving in LPS-inflamed lung from *Mrass et al., 2017*.

For T cells from influenza-infected lung, mice were infected intranasally with $1 \times 10^3$ PFU HKx31 (Charles River). To ensure sedimentation of the virus into the lower respiratory tract, infection was performed while mice were under anesthesia with 90 mg/kg ketamine and 8.1 mg/kg xylazine. In some experiments mice received polyclonal naive GFP+CD8+ T cells from Ubiquitin-GFP animals before infection with influenza. GFP+CD8+ naive T cells were derived from single cell suspensions isolated from spleen and LNs of Ubiquitin-GFP animals, and CD8+ T cells isolated using the CD8a+ T Cell Isolation Kit (Miltenyi Biotec), then transferred via the tail vein into recipient mice. Recipient mice received approximately $10^4$ GFP+CD8+ T cells. All work was done in accordance with approved protocols per IACUC institutional approvals.

For imaging of GFP+CD8+ T cells in influenza-infected lungs, mice were euthanized and lungs from influenza-infected mice were removed at days 7 or 8 post infection. After opening the chest cavity, the lungs were inflated with 2% low melting agarose (Sigma-Aldrich, #A0701) at a temperature of 37°C. We injected one ml of the solution via a catheter through an incision of the trachea. After inflation, the opening in the trachea was sealed with a knot and agarose solidification was induced by exposing the lungs to a phosphate-buffered saline solution with a temperature of 4°C. After harvest, the lungs were transferred into an incubator and transferred within a biosafety cabinet into a POC-R imaging chamber (LaCon). Imaging was performed with a Zeiss LSM800 Airyscan Confocal Microscope. Due to the transparency of the prepared lung tissue, it was possible to visualize at tissue depths of more than 60 µm. *Video 4* shows GFP+CD8+ T cells moving at d8 post infection in influenza-infected lung. In some experiments, imaging was performed with similar setups using a Prairie Ultima Two-photon microscope or a Zeiss LSM 510 microscope. We captured equivalent T cell behavior with the different microscope setups.

Below we summarize the time step sampling and the various metrics used to analyze the T cell motion. These metrics include speed (cell-based and displacement speed), tendency to persist at the same speed, MSD, directionality through the meandering ratio, confined ratio and time, and volume patrolled per time.

### Sampling

Due to the different time steps used in the two-photon microscopy of the different tissues, we sample the position data every 90 s (or as close to 90 s as possible) for each of the tissues to normalize and equalize T cell analyses. Also for tissues which were sampled at a higher frequency, we are able to use all the data for results involving the turning angle by revisiting times that are skipped in an initial 90 s sampling. For example, suppose observations are made at the following times $\{t_0, t_1, t_2, t_3, t_4, ...\}$ where $t_0 = 0$ s, $t_1 = 45$ s, $t_2 = 90$ s, $t_3 = 135$ s, $t_4 = 180$ s and so on. The first sampling of the data retains $\{t_0, t_2, t_4, ...\}$ and the second sampling uses $\{t_1, t_3, t_5, ...\}$. We do not subsample the LPS-inflamed lung data since the time steps are initially 90 s. After this sampling is done, the LN mean time step was 89.9 s with a standard deviation of 2.9 s, the villi mean time step was 93.0 s with a standard deviation of

7.9 s, the lung (Flu) mean time step was 90.0 s with a standard deviation of 0.12 s, and the unsampled lung (LPS) mean time step was 90.0 s with a standard deviation of 0.10 s.

## Speed

If $(x_i, y_i, z_i)$ refers to the position of a cell at time $t_i$ and $(x_{i+1}, y_{i+1}, z_{i+1})$ refers to the position of the cell at time $t_{i+1}$, let $d_{i,i+1}$ represent the distance between the two positions

$$d_{i,i+1} = \sqrt{(x_{i+1} - x_i)^2 + (y_{i+1} - y_i)^2 + (z_{i+1} - z_i)^2}. \tag{1}$$

Two different types of speeds are computed using the positions and times from a T cell: a cell-based speed and displacement-based speed.

### Cell-based speed

If a cell is tracked for $n$ positions and times, the cell-based speed $s_{cell}$ of a cell is computed by summing all the distances traveled by the cell and dividing by the total elapsed time $t_n - t_1$,

$$s_{cell} = \frac{\sum_{i=1}^{n-1} d_{i,i+1}}{t_n - t_1}. \tag{2}$$

### Displacement speed

The displacement speed $s_{displacement}$ is computed using the first and last locations of the cell,

$$s_{displacement} = \frac{d_{1,n}}{t_n - t_1}. \tag{3}$$

### Turning angle and directionality

The turning angle $\theta$ for a T cell given the three positions of the cell {1,2,3} enclosed within circles is shown in *Figure 7A*. If $\mathbf{v}_1$ is the vector formed from positions $(x_1, y_1, z_1)$ and $(x_2, y_2, z_2)$, $\mathbf{v}_1 = (x_2 - x_1, y_2 - y_1, z_2 - z_1)$ and $\mathbf{v}_2$ is the vector formed from positions $(x_2, y_2, z_2)$ and $(x_3, y_3, z_3)$, $\mathbf{v}_2 = (x_3 - x_2, y_3 - y_2, z_3 - z_2)$, the turning angle $\theta$ is computed using

$$\theta = arccos\left(\frac{\mathbf{v}_2 \cdot \mathbf{v}_1}{\|\mathbf{v}_1\|\|\mathbf{v}_2\|}\right). \tag{4}$$

We did not include speeds <1 μm/min as these cells are likely to be considered stopped and turning angles with very small speeds can lead to artifactual angle measurements; for example, small speeds will emphasize turning angles in increments of 45° due to the pixel resolution of the microscope.

Suppose a T cell visits the four locations enclosed within the circles (*Figure 7B*). One can compute the total distance traveled by summing up the distance from location 1 to location 2, $d_{1,2}$, the distance from location 2 to location 3, $d_{2,3}$, and the distance from location 3 to location 4, $d_{3,4}$. The straight line distance can also be computed from the original location 1 to the final location 4, $d_{1,4}$. One measure of a cell's tendency to maintain its direction is the meandering ratio (*Lambert Emo et al., 2016*)

$$M = \frac{d_{1,4}}{d_{1,2} + d_{2,3} + d_{3,4}}.$$

If the ratio $M$ is close to 1, the cell deviates very little from one direction, whereas if $M$ is much <1, the cell moves along a meandering path. In general for $n$ locations, directionality can be measured using the ratio

$$M = \frac{d_{1,n}}{\sum_{i=1}^{n-1} d_{i,i+1}}. \tag{5}$$

## Confined ratio and confined time

We denote the amount of time a T cell lingers in one location as confined time. Given a time $t_i$ and location $(x_i, y_i, z_i)$, we count the time difference between $t_i$ and $t_j > t_i$ as confined time if the difference

$t_j - t_i$ is >150 s and the distance $d_{i,j}$ is <5 μm. Once the cell exits (say at time $t_k$) the 5 μm radius centered about $(x_i, y_i, z_i)$, the cell is tracked anew and the confined time is computed from $t_k$ and location $(x_k, y_k, z_k)$. We call the ratio of confined time to the total time the confined ratio.

In regards to confined time, we calculate the amount of time required to leave a 5-μm radius for each cell position. The time is then averaged over all positions to find the confined time.

## Tendency to persist at a speed

Let $s_b$ and $s_a$ be two consecutive frame speeds (before and after) in $μm/min$ from a T cell track. If $B_i$ represents the event that $s_b$ lies between $i$ $μm/min$ and $(i + 1)$ $μm/min$, then the probability of event $B_i$ occurring is $P(B_i) = m_b/m$, where $m_b$ represents the number of times $i$ $μm/min \leq s_b < (i + 1)$ $μm/min$ and $m$ represents the total number of tracks. If $A_i$ represents the event that $i$ $μm/min \leq s_a < (i + 1)$ $μm/min$, then the probability $P(A_i) = m_a/m$ where $m_a$ represents the number of times $i$ $μm/min \leq s_a < (i + 1)$ $μm/min$. Finally, it follows that $P(A_i \text{ and } B_i) = m_{ab}/m$ where $m_{ab}$ represents the number of times both criteria are satisfied: $i$ $μm/min \leq s_a, s_b < (i + 1)$ $μm/min$ in consecutive frames. According to the definition of conditional probability $P(A_i|B_i) = P(A_i \text{ and } B_i)/P(B_i) = m_{ab}/m_b$. The increased likelihood of persisting at the same speed is then calculated as the ratio $P(A_i|B_i)/P(A_i) = (m \cdot m_{ab})/(m_a \cdot m_b)$.

## Mean squared displacement

Values of the log of the MSD $(d_{1,n})^2$ are plotted against the log of the elapsed time. We limit the elapsed time to 10.5 min. The slope of the linear regression line is computed from the scatter plot and used to characterize the type of motion. Slope values near 1.0 are associated with Brownian motion, values between 1.0 and 2.0 are associated with Lévy walks, and values <1.0 are considered subdiffusive (*Krummel et al., 2016*).

## Rate of volume patrolled

The volume patrolled by a T cell is computed by dividing a 400 μm x 400 μm x 400 μm volume within which a T cell moves into 2.5 μm × 2.5 μm × 2.5 μm cubes. If the distance between a cube center and the T cell center is <5 μm, the cube volume is assumed to be patrolled. We also connect each two successive cell positions with a straight line and assume the cell patrols volume along the straight line. The total volume patrolled is then divided by the time the T cell is tracked.

## Statistical methods

When comparing two groups, p-values were computed using the paired Wilcoxon rank sum test (otherwise known as the Mann–Whitney *U*-test) using the statistical package R with the Bonferroni correction for multiple comparisons.

ANOVA was used to assess both intra- and inter-tissue variabilities. ANOVA uses the *F* distribution which computes a ratio of variability between groups to variability within groups and is commonly used to test differences between more than two groups. The *F* value increases as the means between groups increases and the variability within the groups decreases. The larger the *F* value, the smaller the p-value. To test intra-tissue variability, the groups consisted of frames composed of two-photon tracks within the same tissue of a single mouse. To test inter-tissue variability, the groups consisted of the aggregated frames of two-photon tracks of all mice imaged in the same tissue. These results are discussed in more detail in *Table 2*. We show that while significant variability does exist within the frames of a tissue, the variability can be reduced be eliminating outliers. Our results using the reduced set of files is similar to the complete set of files (for details, see Supplementary Data *Figure 1—figure supplement 3*, *Figure 2—figure supplement 2*, and the associated tables within the figures).

## Acknowledgements

This work was supported by an Institutional Development Award (IDeA) from the National Institute of General Medical Sciences of the National Institutes of Health under Grant P20GM103451 (DJT); NIH 1R01AI097202 (JLC), the Spatiotemporal Modeling Center (P50GM085273), the Center for Evolution and Theoretical Immunology 5P20GM103452 (JLC), AIM CoBRE at UNM HSC NIH P20GM121176 (JLC; PM), NIH 5 T32 AI007538-19 (JRB), institutional support from dedicated health research funds from UNM SOM (JLC), Women in Science Award from UNM, DARPA/AFRL FA8650-18-C-6898 (JLC)

and in part institutional support from Dedicated Health Research Funds from the UNM School of Medicine (JLC and PM); and support by the University of New Mexico Comprehensive Cancer Center Support Grant NCI P30CA118100 (JLC and PM).

## Additional information

### Funding

| Funder | Grant reference number | Author |
| --- | --- | --- |
| National Institutes of Health | P20GM103451 | David J Torres |
| National Institutes of Health | 1R01AI097202 | Judy L Cannon |
| National Institutes of Health | P50 GM085273 | Judy L Cannon |
| National Institutes of Health | 5P20GM103452 | Judy L Cannon |
| National Institutes of Health | P20GM121176 | Judy L Cannon Paulus Mrass |
| National Institutes of Health | 5 T32 AI007538-19 | Janie Byrum |
| University of New Mexico | School of Medicine | Judy L Cannon Paulus Mrass |
| University of New Mexico | DARPA/AFRL FA8650-18-C-6898 | Judy L Cannon |
| University of New Mexico | NCI P30CA118100 | Judy L Cannon Paulus Mrass |

The funders had no role in study design, data collection, and interpretation, or the decision to submit the work for publication.

### Author contributions

David J Torres, Conceptualization, Software, Formal analysis, Funding acquisition, Visualization, Writing - original draft, Writing - review and editing; Paulus Mrass, Resources, Data curation, Methodology; Janie Byrum, Data curation, Methodology; Arrick Gonzales, Dominick N Martinez, Evelyn Juarez, Software; Emily Thompson, Vaiva Vezys, Data curation; Melanie E Moses, Conceptualization, Methodology; Judy L Cannon, Conceptualization, Data curation, Formal analysis, Funding acquisition, Writing - original draft, Project administration, Writing - review and editing

### Author ORCIDs

David J Torres (iD) http://orcid.org/0000-0003-2469-5284
Judy L Cannon (iD) http://orcid.org/0000-0003-0069-8106

### Ethics

All work was done in accordance with approved protocols per IACUC institutional approvals, IACUC Animal approval #: 21-201165-HS.

### Decision letter and Author response

Decision letter https://doi.org/10.7554/eLife.84916.sa1
Author response https://doi.org/10.7554/eLife.84916.sa2

## Additional files

### Supplementary files

- MDAR checklist

• Supplementary file 1. Supplementary T cell files.

### Data availability

Datasets are attached as Supplementary Materials. The code used for analysis can be downloaded at: https://github.com/davytorres/T-cell-analysis-tool (copy archived at *Torres, 2023*).

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
