## [Editor Report]

Your unbiased analysis of T cell motion in different tissue settings has revealed that many parameters of T cell motion are similar across tissues, whereas other parameters, particularly related to confinement, are highly sensitive to the tissue microenvironment. Your agnostic approach, meticulous analysis, and clear results are valuable and the technical approach is convincing in its simplicity. The work will be of interest to scientists working in diverse fields from immunologists tracking white blood cell motion to developmental biologists tracking germ cells in *Drosophila* larvae.

---

## [Decision Letter]

**Decision letter after peer review:**

Thank you for submitting your article "Quantitative analyses of T cell motion in tissue reveals factors driving T cell search in tissues" for consideration by *eLife*. Your article has been reviewed by 3 peer reviewers, including Michael L Dustin as Reviewing Editor and Reviewer #1 and the evaluation has been overseen by Satyajit Rath as the Senior Editor. The following individual involved in the review of your submission has agreed to reveal their identity: Emmanuel Donnadieu (Reviewer #2).

Essential revisions:

The reviewers agree that the analysis performed is valuable and solid. The importance of the work can be increased by addressing specific suggestions of the reviewers that may require the introduction of additional data that hopefully was already collected.

1) All reviewers wondered about the microanatomical differences in the lung, villi, and lymph nodes. If second harmonic signals were collected then including some of this data would be a helpful start and such data was often collected alongside fluorescence data.

2) There were also questions about the statistical analysis that should be straightforward to address given the authors' background.

3) There was a concern about having the same T cells in each compartment. This may be difficult to achieve, but perhaps the phenotype of T cells in each compartment could be assessed for relevant parameters, or perhaps such data already exists in the literature- for example, levels of CD11a, CD103, CD44, CD62L, CCR7 and CXCR4, for example, could be investigated for T cells in each site to allow potentially informative comparisons.

*Reviewer #1 (Recommendations for the authors):*

The turning angle distribution in the lymph node has a bias toward small turning angles and thus displays features of a correlated random walk as previously described by this group and others. It is interesting that the turning angle distribution is flatter in the lung, particularly the LPS-treated lung, which is more similar to Brownian motion in this respect. The authors seem to reject this, but this aspect was striking and surprising.

Movement of the T cells in the lymph node and villi is likely scaffolded on stromal cells. Is this also true in the lung or is the connective tissue more similar to a dense collagen gel? Was second harmonic data collected in each study or any other analysis of the tissue context that might help understand the movement pattern in the lung? Even if this cannot be fully explored, it could provide a direction for future research to provide some examples, if available.

Some of the p-values from the Mann-Whitney type test suggest a very high likelihood that the behaviours are different across sites, but some of this may be due to the very large N from the analysis of individual T cells. It's not clear that these are really independent measurements as multiple T cells from one field will share microanatomical niches and may also share technical artefacts such as drift. Another way to look at these data are to bin the values by field or experimental days and retest if the mean values are still significantly different across sites. If the values are still significant when values from different animals/preparations are compared, then the p values will likely be more modest, but there would be greater confidence that the values are independent.

*Reviewer #2 (Recommendations for the authors):*

1) This study would have been even more important if the authors had provided information on the reasons why T-cell migration in the lung is so different than in other organs. For instance, the back-and-forth migration observed in the lung is striking. Can the authors try to correlate this feature with some external determinants of the lung tissue? It can be assumed that this type of motion is dependent on structural elements (vessels, matrix fibers) that constrain T cells forcing them to migrate back and forth. In a previous study (PMID: 29044117), the authors immunostained the lung vasculature and thereafter monitor T cells in relation to vessels. Can they use such methods together with second harmonic generation microscopy to identify external factors that can possibly control T cell migration in the lung? This would strengthen the impact of the study.

2) The persistence plot shown in figure 1C is interesting as such analysis has rarely been done. However, I am not entirely convinced by some claims of the authors. They conclude that T cells with the lowest and highest speed persist in moving at the same speed. It is likely that T cells at the lowest speeds are arrested and remain arrested during the recording. However, if this first point (T cells with the lowest speed) is removed from the graph the correlation is not evident. This should be commented on.

3) The data presented are mostly histograms comparing the motility parameters of T cells in different tissues. The study would benefit from including videos and illustrations from videos (e.g., tracks with x and y coordinates). These will help to convey some of the critical aspects reported in this study.

*Reviewer #3 (Recommendations for the authors):*

Specific corrections to the text and figures are required:

1) There is insufficient data in Table 1 to allow the reader to understand the differences between the different models. The simple naïve versus activated designation is misleading and fails to convey the distinct differences in the experimental design of each group. Additional columns should include: 1) mode of activation e.g., in vitro versus in vivo; 2) specificity e.g., polyclonal versus monoclonal; and 3) in situ imaging e.g., intravital versus tissue explant.

2) There is no reference to Figure 1 in the text. Data start at Figure 2.

3) More information is needed as to the micro-anatomical location of the imaging volumes in the two lung models – the lung is a large organ, so specifying where the imaging volumes were taken in the two models (e.g., depth, lobe location, distance from the epithelium of the airway or bronchi, etc.). It will be important to determine if motility differences in the lung could be attributed to a difference in the micro-anatomical position chosen for image acquisition.

---

## [Author Response]

Essential revisions:The reviewers agree that the analysis performed is valuable and solid. The importance of the work can be increased by addressing specific suggestions of the reviewers that may require the introduction of additional data that hopefully was already collected.

We thank the reviewers for their insightful and useful comments to improve the manuscript. We have addressed all their comments below and hope that the reviewers and editor agree that the manuscript is now suitable for publication in *eLife*.

1) All reviewers wondered about the microanatomical differences in the lung, villi, and lymph nodes. If second harmonic signals were collected then including some of this data would be a helpful start and such data was often collected alongside fluorescence data.

While we agree that this would be the important and logical next step to this work, the data to specify microanatomical differences between lung, villi, and lymph nodes is not available and we were not able to complete additional image collection in a timely manner to include with the current revision. In addition, part of the reason we performed the analysis in this manuscript is to NOT consider the microanatomical differences within tissue but to compare across tissues. Significant molecular and cellular differences across tissues are well documented, including differences in extracellular matrix composition, chemokine expression, and cell types, just to name a few. It is likely that multiple cell types and molecules contribute to potential differences in T cell motility, and it was not possible to test individual molecules’ contribution to motility changes. We compared motility patterns rather than the effect of individual molecular effects on motility to be able to compare across tissues. We purposefully did NOT choose specific regions within any tissue other than areas infiltrated by T cells. This unbiased approach allowed us to compare how T cells move in different tissues without selecting for any specific regions or potentially biasing towards specific types of interactions.

We use quantitative motility parameters that are independent of molecular factors in order to compare across tissues. The question of how specific molecular and physical interactions within different tissues affect motility remains a question for additional work outside the scope of this manuscript. In fact, different microanatomical features and multiple molecules likely all contribute to the differences in motility observed in T cell motion. As our overarching question remained to identify whether distinct tissues result in any differences in T cell motion, we specifically analyzed all T cells without “selecting” specific areas based on microanatomical differences.

In response to concerns by both reviewers 1 and 2 regarding intra-tissue variability, we have now performed an additional analysis shown in Table 2. Our new analysis with ANOVA shows that within tissue motility can be highly statistically significantly different (see Column 3, Rows 3-6 of Table 2). The most significant variation within tissue is in the lymph nodes, but all tissues show significant differences within tissue. These new analyses suggest that microanatomical areas within tissues do indeed mediate different types of T cell motion.

Importantly, our new analyses show that even if we remove the statistically significantly different frames from the overall sample, the motility patterns of the remaining cells within any one tissue are similar to the motility patterns with all the cells included in the original analyses. We now show the new analyses with statistically significantly different frames removed as a comparison in Supplementary Data (Figure 1 – supplement 3, Figures 2-6 supplement 2 and associated tables). A detailed description of the comparison between the full dataset (Figures 1-6 and Tables 3-11) with the reduced dataset (Figure 1 – supplement 3, Figures 2-6 supplement 2, and associated tables) is now included in the “Response to multiple reviewers” on page 5 of this file. We have added a discussion of this point to the manuscript in the new Section 4.8 within the Materials and methods and on page 5 of Results and page 16 in the Discussion. Due to the desire to not eliminate intra-tissue variation, we have not removed cells and frames from our analyses in the main manuscript. We have kept all the frames as we have no scientific reason to exclude their analyses as we believe intra-tissue variability is important to consider and include in the analysis.

2) There were also questions about the statistical analysis that should be straightforward to address given the authors' background.

We performed an Analysis of Variance (ANOVA) study which computes a ratio of variance using a ratio of differences in means between tissues and within tissues. The ANOVA test uses a F-distribution which computes a ratio of between-groups variance and within-groups variance. The larger the differences from the mean between tissues and the smaller the differences in the means within tissues, the larger the F-value and the smaller the p-value. The p-values are shown in Table 2. We have added an additional section to the Materials and methods, Section 4.8 Statistical Methods as well as a more extensive explanation in the caption of Table 2.

3) There was a concern about having the same T cells in each compartment. This may be difficult to achieve, but perhaps the phenotype of T cells in each compartment could be assessed for relevant parameters, or perhaps such data already exists in the literature- for example, levels of CD11a, CD103, CD44, CD62L, CCR7 and CXCR4, for example, could be investigated for T cells in each site to allow potentially informative comparisons.

The reviewers are correct to point out that we expect differences between T cells in each compartment due to migration properties. Expression of all the markers specified are different in each tissue. The CD8 T cells used in each model have previously been described in detail and form well-established models for T cell activation and T cell function. For example, chemokine receptors and some integrins are differentially expressed in order to mediate migration into different tissues; for example, CCR7 is critical for homing to lymph nodes ([2],[3],[4]) while CXCR3 and other chemokine receptors are important for lung homing ([5], [6], [7], [8], [9]). We have confirmed these findings and find that naive T cells in LNs have high CCR7, high CD62L, and low CD44 as previously shown (Author response image 1).

**Author response image 1. sa2fig1:** Characterization of naive CD4 and CD8 cells. GFP+ T cells were isolated from LNs and spleen and naïve C57/BI/6 animals and stained with indicated markers and analyzed by flow cytometry. Majority of naïve CD4 and CD8 T cells are CD62LhiCD44lolL7R+CCR7+.

P14 CD8 T cells activated in the LCMV model have been extensively characterized and published, with multiple publications showing expression of canonical activation markers such as CD44 as well as gut specific homing markers including CCR9 and a4b7 integrins ([10], [11], [12]). The data used here has also been previously published demonstrating expression of CD69 and CD103 [1]. Because characterization of these T cells has been published, we do not reproduce the activation markers here.

CD8 T cells in LPS-inflamed lung were activated in vitro by anti-CD3/anti-CD28, then adoptively transferred into animals treated with LPS intranasally. T cells isolated from the lung were analyzed by flow cytometry after adoptive transfer and uniformly CD44hiCD62Llo (Author response image 2). CD8 T cells responding to influenza infection at d8 post infection in the lung express CD44hi as well as CXCR4 with very little CXCR3 at d8 (Author response image 3).

**Author response image 2. sa2fig2:** Characterization of activated CD8 T cells in vitro activated by anti-CD3 and anti-CD28, then adoptively transferred into LPS inflamed lung. CD8 T cells from Ub-GFP animals were isolated by Miltenyi selection, then adoptively transferred into mice that had received LPD intranasally the prior. At D7, lungs were removed and single cells suspension prepared, and CD8^+^GFP+ cells were analyzed for CD44 and CD62L. Adoptively transferred CD8 T cells were predominantly CD44hiCD62Llo.

**Author response image 3. sa2fig3:** Characterization of activated CD8 T cells activated in lung of influenza infected animals at d8 post infection. C57Bl/6 animals were infected with 1x10^3^ HKx31 and at d8 post infection, lungs were removed and single cell suspension prepared from lung tissue and analyzed by flow cytometry. CD8 T cells were CD44hi and some fraction CXCR4+ but very few CXCR3+.

The list of markers from reviewers and editors are only some of the many likely differences between cell types and the tissues studied in the manuscript. The differences in these and many other markers are well established and not novel, so we do not include these panels in the manuscript and include them here for reviewers. In this manuscript, we establish differences in motility patterns. Testing the specific role of individual molecules on T cell motility is well beyond the scope of this manuscript. In fact, it is likely that a combination of multiple interactions as well as structural features of tissues contribute to the motility differences we observe, and we now discuss this in more detail in the revised manuscript in both the introduction, results, and discussion.

Response to multiple reviewers: Both reviewers 1 and 2 brought up the possibility that within tissue differences may account for significant variability and asked for comparison of within tissue versus between tissue variability.

We performed an ANOVA analysis of each cell-based speeds within tissue (see column 3 of Table 2). Imaging of each individual tissue includes capturing data from multiple animals on different days with different frames. ANOVA analysis shows that within the same tissue, there are highly statistically significant differences.

Our analysis shows that “within” tissue variability Table 2 (Rows 3-6, Column 3) could exceed “between” tissue variability, for example, “within” LN variability p-value was p=1.7 x 10^-204^ which exceeded “between” tissue variability with p=3.5 x 10^-11^ (Row 2, Column 3).

To decrease the “within” tissue variability, we identified the most variable fields within each tissue and removed those fields. The “within” tissue variability and “between” tissue variability was then recalculated (Table 2, Column 5). Removing the outlier frames significantly decreased the “within” tissue variability to p=4.8 x 10^-5^ in LNs, demonstrating that these removed frames were outliers compared with the data from the same tissue.

The result of eliminating the cells from frames that were statistical outliers resulted in a reduced set of files from the same tissue. 19 frames (1659 cells) of the original 40 frames (4400 cells) were retained for the LN; 8 frames (296 cells) were retained of the original 10 frames (425 cells) for the villi; and 4 frames (297 cells) were retained of the original 5 frames (355 cells) for the Lung (flu). No frames were removed from the Lung (LPS) because no frame was shown to be statistically significantly different from the others. Removal of outlier frames resulted in a large increase in the p-value within tissue and decrease in p-value between tissue (for example, compare Table 2 Column 3 and Column 5; within LN p=4.8 x 10^-5^ while between tissue p=2.1 x 10^-20^). The cell numbers reported in Table 1 in the original submission were also adjusted to account for cells that were excluded from analysis because they were not tracked long enough (e.g. shorter than 150 seconds).

We then re-analyzed all the intra-tissue comparisons for the major motility parameters, including key parameters of cell-based speed, displacement speed, volume per time, and meandering ratio p-values. Figures in the revised manuscript Supplementary Data, Figure 1 – supplement 3, Figures 2-6 – supplement 2 with their accompanying tables show statistical differences between tissues after removal of outlier frames. All motility analyses were performed on the reduced data sets in the Supplementary Data.

Interestingly, very few between tissue motility parameters changed with the outlier frames removed (compare main manuscript Figures 1-6 with complete dataset; supplemental Figures 1 – supplement 3, Figures 2-6 – supplement 2 with the reduced dataset). The largest changes were in cell-based speed and displacement speed shown in Figure 1 and Figure 1 – supplement 3. For example, the cell-based median speed of T cells in LN increased from 6.2 um/min (all frames, Figure 1A) to 7.2 um/min (reduced frames, Figure 1 – supplement 3A). However, the conclusion that T cells in the LN and villi moved with similar speeds while T cells in the lung moved more slowly remains the same. The same trends remained the same for all of the motility parameters analyzed between the complete and reduced datasets.

The p-values in both sets are also similar when comparing cell-based speed across tissues. We note a few differences. With the full dataset, there is no statistical difference between cell-based speed between T cells in LN, villi and influenza-infected lung while with the reduced dataset, the influenza infected lung is slightly slower than the villi (Table 3 vs Figure 1D – supplement 3). The same was seen for turning angles between LN and villi (Table 6 vs Figure 3C – supplement 2) and confined time (Table 9 vs Figure 5D – supplement 2). MSD showed no difference between LN and villi in the full dataset while the villi showed significantly lower MSD in the reduced dataset (Table 5 vs Figure 2C – supplement 2). The full dataset also showed no statistical difference between the MSD of T cells from influenza-infected lung versus the LPS inflamed lung while the reduced dataset did show statistically significant difference (Table 5 vs Figure 2C – supplement 2).

Since there is no scientific reason to exclude the outlier frames (as stated in the response to main critiques above) and because the main conclusions do not change for between tissue comparisons, we have included all the data in the main text, while including the analysis of the reduced dataset in the Supplementary Data section.

Reviewer #1 (Recommendations for the authors):The turning angle distribution in the lymph node has a bias toward small turning angles and thus displays features of a correlated random walk as previously described by this group and others. It is interesting that the turning angle distribution is flatter in the lung, particularly the LPS-treated lung, which is more similar to Brownian motion in this respect. The authors seem to reject this, but this aspect was striking and surprising.

While the turning angle distribution in the LPS-treated lung is flatter, it still can be considered bimodal. If the motion was Brownian, the turning angle distribution should be uniform. We have now added a sentence to discuss this point on page 9 (Section 2.4) and page 16 of the Discussion.

Movement of the T cells in the lymph node and villi is likely scaffolded on stromal cells. Is this also true in the lung or is the connective tissue more similar to a dense collagen gel? Was second harmonic data collected in each study or any other analysis of the tissue context that might help understand the movement pattern in the lung? Even if this cannot be fully explored, it could provide a direction for future research to provide some examples, if available.

We previously published that T cells move along vasculature in the lung in LPS-inflamed lungs [13]. We did not collect second harmonic data in the lung. We agree with the reviewer that a more detailed analysis in each individual tissue of the types of cells and structures that are likely to “guide” T cell movement will be important for future studies.

Some of the p-values from the Mann-Whitney type test suggest a very high likelihood that the behaviours are different across sites, but some of this may be due to the very large N from the analysis of individual T cells. it's not clear that these are really independent measurements as multiple T cells from one field will share microanatomical niches and may also share technical artefacts such as drift. Another way to look at these data are to bin the values by field or experimental days and retest if the mean values are still significantly different across sites. If the values are still significant when values from different animals/preparations are compared then the p values will likely be more modest, but there would be greater confidence that the values are independent.

This was discussed above at the end of the response to essential revisions requested by the editor as well as response to multiple reviewers. Please see above for a detailed discussion of how we tested for within tissue differences and the rationale for keeping all data within the dataset.

Reviewer #2 (Recommendations for the authors):1) This study would have been even more important if the authors had provided information on the reasons why T-cell migration in the lung is so different than in other organs. For instance, the back-and-forth migration observed in the lung is striking. Can the authors try to correlate this feature with some external determinants of the lung tissue? It can be assumed that this type of motion is dependent on structural elements (vessels, matrix fibers) that constrain T cells forcing them to migrate back and forth. In a previous study (PMID: 29044117), the authors immunostained the lung vasculature and thereafter monitor T cells in relation to vessels. Can they use such methods together with second harmonic generation microscopy to identify external factors that can possibly control T cell migration in the lung? This would strengthen the impact of the study.

In response to point #1 of reviewer #1, we have now added more discussion of potential differences in molecules, cells, and structures including mechanical tension and next steps in the introduction and discussion.

2) The persistence plot shown in figure 1C is interesting as such analysis has rarely been done. However, I am not entirely convinced by some claims of the authors. They conclude that T cells with the lowest and highest speed persist in moving at the same speed. It is likely that T cells at the lowest speeds are arrested and remain arrested during the recording. However, if this first point (T cells with the lowest speed) is removed from the graph the correlation is not evident. This should be commented on.

To address this concern, we increased the number of speed bins on the horizontal axis for speeds up to 3 um/min and assessed persistence (see Author response image 4). We find that there is still a slight increased probability for T cells to move persistently at lower speeds even for T cells moving at < 0.75 um/min. We now include a more detailed discussion of persistence analysis in Section 2.2 on pages 7 and 8.

**Author response image 4. sa2fig4:** Persistance plot at low speeds.

3) The data presented are mostly histograms comparing the motility parameters of T cells in different tissues. The study would benefit from including videos and illustrations from videos (e.g., tracks with x and y coordinates). These will help to convey some of the critical aspects reported in this study.

We have now added videos for movement in each tissue. Some have been previously published (videos from naive T cells in LN published in Fricke et al. 2016 Video 1 [14]; d8 villi from Thompson et al. 2019 [1], Video 2; d8 from LPS-inflamed lung from Mrass et al. 2017 [13], Video 3). Videos from these publications have been reproduced with appropriate attribution. Video 4 from T cells on d8 post infection in influenza-infected lung is also included.

Reviewer #3 (Recommendations for the authors):Specific corrections to the text and figures are required:1) There is insufficient data in Table 1 to allow the reader to understand the differences between the different models. The simple naïve versus activated designation is misleading and fails to convey the distinct differences in the experimental design of each group. Additional columns should include: 1) mode of activation e.g., in vitro versus in vivo; 2) specificity e.g., polyclonal versus monoclonal; and 3) in situ imaging e.g., intravital versus tissue explant.

We have now added greater detail to both methods (Revised Table 1) as well as basic characterization of cells (described above in the response). We have also added a comparison of d5 and d8 activated CD8 T cells in the villi (see Supplementary Data).

2) There is no reference to Figure 1 in the text. Data start at Figure 2.

We have now added a reference to Figure 1 in the methods.

3) More information is needed as to the micro-anatomical location of the imaging volumes in the two lung models – the lung is a large organ, so specifying where the imaging volumes were taken in the two models (e.g., depth, lobe location, distance from the epithelium of the airway or bronchi, etc.). It will be important to determine if motility differences in the lung could be attributed to a difference in the micro-anatomical position chosen for image acquisition.

We discuss this at great length above in the response to the editor. We captured images at multiple depths with the limitation being the depth of the two-photon laser penetration into tissue. However, we performed no “selection” based on “location” within tissue other than where CD8 T cells localize. Microanatomical differences clearly play a role in tissue motility but the overall motility characteristics remain the same despite removing fields from each dataset.

References

[1] Thompson EA, Mitchell JS, Beura LK, Torres DJ, Mrass P, Pierson MJ, Cannon JL, Masopust D, Fife BT, Vezys V (2019) Interstitial Migration of CD8abT cells in the small intestine is dynamic and is dictated by environmental cues. Cell Reports. 26:2859-2867. https://doi.org/10.1016/j.celrep.2019.02.034.

[2] Förster R, Davalos-Misslitz AC, Rot A (2008) CCR7 and its ligands: balancing immunity and tolerance. Nat Rev Immunol. 8(5):362-71. doi: 10.1038/nri2297.

[3] Asperti-Boursin F, Real E, Bismuth G, Trautmann A, Donnadieu E (2007) CCR7 ligands control basal T cell motility within lymph node slices in a phosphoinositide 3-kinase-independent manner. Journal of Experimental Medicine. 204(5):1167-1179. https://doi: 10.1084/jem.20062079.

[4] Worbs T, Mempel TR, Bölter J, von Andrian UH, Förster R (2007) CCR7 ligands stimulate the intranodal motility of T lymphocytes in vivo. J Exp Med. 204(3):489-95. doi: 10.1084/jem.20061706.

[5] Ozga AJ, Chow MT, Lopes ME, Servis RL, Di Pilato M, Dehio P, Lian J, Mempel TR, Luster AD (2022) CXCL10 chemokine regulates heterogeneity of the CD8^+^ T cell response and viral set point during chronic infection. Immunity. 55(1):82-97. https://doi: 10.1016/j.immuni.2021.11.002.

[6] Kohlmeier JE, Cookenham T, Miller SC, Roberts AD, Christensen JP, Thomsen AR, Woodland DL (2009) CXCR3 directs antigen-specific effector CD4^+^ T cell migration to the lung during parainfluenza virus infection. J Immunol. 183(7):4378-84. https://doi: 10.4049/jimmunol.0902022.

[7] Fadel SA, Bromley SK, Medoff BD, Luster AD (2008) CXCR3-deficiency protects influenza-infected CCR5-deficient mice from mortality. Eur J Immunol. 38(12):3376-87. https://doi: 10.1002/eji.200838628.

[8] Wein AN, McMaster SR, Takamura S, Dunbar PR, Cartwright EK, Hayward SL, McManus DT, Shimaoka T, Ueha S, Tsukui T, Masumoto T, Kurachi M, Matsushima K, Kohlmeier JE (2019) CXCR6 regulates localization of tissue-resident memory CD8 T cells to the airways. J Exp Med. 216(12):2748-2762. https://doi: 10.1084/jem.20181308.

[9] Mikhak Z, Strassner JP, Luster AD (2013) Lung dendritic cells imprint T cell lung homing and promote lung immunity through the chemokine receptor CCR4. J Exp Med. 210(9):1855-69. https://doi: 10.1084/jem.20130091.

[10] Masopust D, Choo D, Vezys V, Wherry EJ, Duraiswamy J, Akondy R, Wang J, Casey KA, Barber DL, Kawamura KS, Fraser KA, Webby RJ, Brinkmann V, Butcher EC, Newell KA, Ahmed R (2010) Dynamic T cell migration program provides resident memory within intestinal epithelium. J Exp Med. 207(3):553-64. https://doi: 10.1084/jem.20090858.

[11] Olson MR, McDermott DS, Varga SM (2012) The initial draining lymph node primes the bulk of the CD8 T cell response and influences memory T cell trafficking after a systemic viral infection. PLoS Pathog. 8(12):e1003054. https://doi: 10.1371/journal.ppat.1003054.

[12] Masopust D, Murali-Krishna K, Ahmed R (2007) Quantitating the magnitude of the lymphocytic choriomeningitis virus-specific CD8 T-cell response: it is even bigger than we thought. J Virol. 81(4):2002-11. https://doi: 10.1128/JVI.01459-06.

[13] Mrass P, Oruganti SR, Fricke GM, Tafoya J, Byrum JR, Yang L, Hamilton SL, Miller MJ, Moses ME, Cannon JL (2017) ROCK regulates the intermittent mode of interstitial T cell migration in inflamed lungs. Nature Communications. 8(1):1010. https://doi: 10.1038/s41467-107-010

[14] Fricke GM, Letendre KA, Moses ME, Cannon JL (2016) Persistence and adaptation in immunity: T cells balance the extent and thoroughness of search. PLoS Comput Biol. 12(3):e1004818. https://doi:10.1371/journal.pcbi.1004818.32-2.

[15] Paulus M, Ichiko K, Byrum JR, Torres D, Baker SF, Cannon JL (2022) CXCR4 controls movement and degranulation of CD8^+^ T cells in the influenza-infected lung via differential effects on interaction and tissue scanning. bioRxiv. https://doi.org/10.1101/2022.09.13.507813.